# TangleScore: Tangle-Guided Purge and Imprint for Unstructured Knowledge Editing

**Hao-Xiang Xu**[1,2]\*, **Ziqi Peng**[1]\*, **Jun-Yu Ma**[1,2], **Yuhao Sun**[1], **Zhen-Hua Ling**[1,2], **Jia-Chen Gu**[3]†

[1]University of Science and Technology of China
[2]National Engineering Research Center of Speech and Language Information Processing
[3]University of California, Los Angeles
{nh2001620,aisis,mjy1999,syh3327}@mail.ustc.edu.cn,
zhling@ustc.edu.cn, gujc@ucla.edu

## Abstract

Large language models (LLMs) struggle with inaccurate and outdated information, driving the emergence of *knowledge editing* as a lightweight alternative. Despite their effectiveness in modifying structured knowledge, existing editing methods often fail to generalize to unstructured cases, particularly those involving inherently *hard-to-edit* knowledge, where the original facts tend to be more resistant to change. To address this, we propose a metric, TangleScore, that quantifies the intrinsic difficulty of editing a given knowledge instance. This difficulty, in turn, strongly correlates with the model's ability to generalize the edit to paraphrased and related prompts. Building on this insight, we introduce a TangleScore-driven method termed **P**urge-**I**mprint **P**atch **E**diting (PIPE), an editing framework that adaptively modulates the purge and imprint of knowledge based on TangleScore of the target knowledge to be edited, thus adjusting the editing strength to match the instance's difficulty, thereby enabling more precise and effective model updates. Experiments applying PIPE to four LLMs of varying sizes on two unstructured knowledge editing datasets show that PIPE significantly outperforms previous editing methods by 6.49% in terms of generalization performance. Extensive evaluation show that PIPE also exhibits effectiveness in structured knowledge editing and strong robustness under batch and sequential editing. The code is available at https://github.com/famoustourist/TangleScore.

## 1 Introduction

Despite the remarkable capabilities of large language models (LLMs) (OpenAI, 2023; Dubey et al., 2024), they often exhibit hallucinations due to incorrect or outdated knowledge embedded in their parameters (Zhang et al., 2023; Ji et al., 2023). Given the high cost of retraining, there has been increasing interest in *knowledge editing* (Sinitsin et al., 2020; Zhu et al., 2020; Dai et al., 2022; Mitchell et al., 2022b; Meng et al., 2022; 2023; Wang et al., 2024; Ma et al., 2024; Gu et al., 2024; Fang et al., 2025; Xu et al., 2025), which aims to revise knowledge in LLMs efficiently. However, most existing methods focus on structured knowledge, while approximately 80% of real-world knowledge is unstructured (Bavota, 2016), as illustrated in Figure 1(a). This mismatch limits their applicability in broader scenarios. Although recent studies have attempted to improve editing in unstructured contexts (Deng et al., 2025; Jiang et al., 2025), a clear gap remains between accuracy and generalization, making unstructured knowledge editing a significant challenge for LLMs.

To diagnose this discrepancy, we conduct a knowledge-wise analysis of the editing outcomes across existing methodologies. Our findings reveal that while these methods demonstrate balanced performance in terms of accuracy and generalization for certain types of knowledge, they exhibit significant degradation when applied to specific samples. Besides, such degradation is not randomly distributed but follows consistent patterns across different editing methods and diverse models. We hypothesize that this stems from a misalignment between the intensity of editing perturbations and

---

\*Equal contribution.
†Corresponding author.

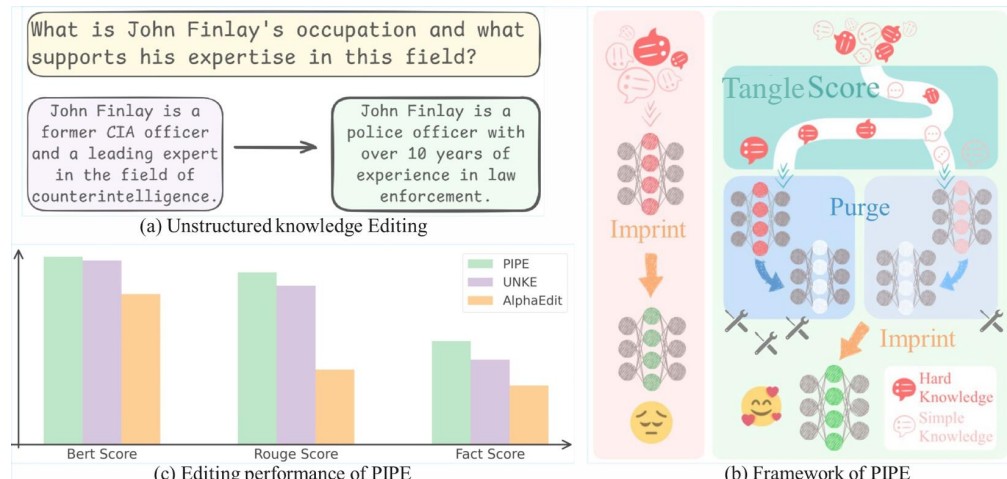

Figure 1: (a) An example of unstructured knowledge editing. (b) A comparison between PIPE and current editing methods. PIPE contains a "purge" phase that utilizes TANGLESCORE to quantify knowledge difficulty, applying different purge rates based on their TANGLESCORE values. (c) A comparison of editing performance between PIPE and current editing methods.

the degree of coupling between the target knowledge and the model: current methods often fail to effectively overwrite the model's internal reliance on outdated knowledge, especially when the model's deeply embedded prior knowledge strongly resists modification.

To identify these knowledge instances, we propose a metric termed TANGLESCORE, designed to measure the degree of entanglement between the to-be-edited knowledge and the model's existing knowledge. This metric jointly considers the discrepancy between model responses to pre-edited and target knowledge, along with their semantic similarity. TANGLESCORE is an intrinsic property determined solely by the base model and the knowledge instance, independent of the editing methods used. Notably, TANGLESCORE strongly correlates with the generalization performance of editing operations, effectively identifying cases where the model may memorize an edit yet still fail to answer related paraphrases or follow-up questions correctly. By leveraging TANGLESCORE, editing methods can adapt to the varying difficulty of knowledge instances, enabling more targeted intervention on hard cases and substantially improving editing effectiveness where previous methods tend to fail.

Consequently, we propose **P**urge-**I**mprint **P**atch **E**diting (PIPE), a method that can adaptively adjust the purge strategy based on the difficulty of the knowledge, applying more aggressive forgetting for hard-to-edit knowledge. As shown in Figure 1(b), PIPE adopts a two-stage editing paradigm. PIPE first introduces a knowledge purge function to actively suppress the influence of conflicting original knowledge. It dynamically adjusts the degree of purge based on the difficulty of the target knowledge, promoting deeper forgetting for harder edits to minimize interference. Next, PIPE imprints new knowledge using a cause-driven optimization strategy, directly modifying the final token of the input through a dedicated knowledge imprint function. This localized editing improves precision without affecting the overall behavior of the model. As depicted in Figure 1(c), these two stages enable PIPE to separate purging from imprinting, enhancing its robustness to prior knowledge interference.

In order to validate the effectiveness of the proposed PIPE, we conduct a comprehensive evaluation using state-of-the-art LLMs of varying sizes, such as LLaMA3-8B-Instruct (Dubey et al., 2024), LLaMA2-7B-Chat (Touvron et al., 2023), LLaMA2-13B-Chat (Touvron et al., 2023) and Qwen2.5-7B-Instruct (Yang et al., 2024). The UNKEBench (Deng et al., 2025) and AKEW (Wu et al., 2024) are selected to demonstrate the generalizability of our approach across different knowledge formats. Experimental results show that PIPE improves generalization by 6.49% in knowledge editing tasks, compared to the state-of-the-art methods. We also evaluate PIPE on the traditional structured knowledge editing benchmark KEBench (Wei et al., 2024), and conduct both batch and sequential editing, demonstrating the strong robustness of our approach across various settings.

In summary, our contributions in this paper are three-fold: (1) This paper presents the TANGLESCORE to quantify the difficulty of editing certain knowledge and demonstrates that prior methods are less

effective at samples with higher difficulty. (2) A method termed PIPE is proposed, which aims to improve the model's ability to edit hard-to-edit unstructured knowledge by explicitly introducing a purge phase before imprinting new factual information. (3) Evaluation results show that PIPE enhances both the effectiveness and generalization of unstructured knowledge editing.

## 2 RELATED WORK

**Knowledge Editing**   Conventional knowledge editing methods either modify parameters or preserve them via auxiliary components. *Parameter-Modifying Methods* directly update weights to encode new knowledge. Meta-learning approaches such as KE (Cao et al., 2021), MEND (Mitchell et al., 2022a), and InstructEdit (Zhang et al., 2024) use hypernetworks, with MEND employing low-rank decomposition. Locate-then-edit methods such as ROME (Meng et al., 2022), MEMIT (Meng et al., 2023), and EAC (Xu et al., 2025) apply causal tracing and compress edits to reduce forgetting, while PRUNE (Ma et al., 2025) restricts conditional branches. *Parameter-Preserving Methods* retain original weights via external modules. ICE (Zheng et al., 2023) and DeCK (Bi et al., 2024) use in-context learning; SERAC (Mitchell et al., 2022b) uses external memory; T-Patcher (Huang et al., 2023) and CaliNet (Dong et al., 2022) add neurons; GRACE (Hartvigsen et al., 2023) replaces hidden states with codebooks; WISE (Wang et al., 2024) uses parameterized memory for integration.

**Unstructured Knowledge Editing**   Recent work extends knowledge editing to unstructured knowledge in free-form text, beyond structured triples. Wu et al. (2024) point out the limitations of prior evaluation protocols and introduce AKEW, a benchmark for unstructured editing. To improve editing in this setting, Deng et al. (2025) propose UNKE, which updates all parameters in a single layer to improve knowledge absorption, together with UNKEBench for evaluation. Huang et al. (2024) further propose a dynamic perception module that efficiently identifies commonsense-relevant parameters for more accurate updates. Jiang et al. (2025) propose AnyEdit to decompose long-form knowledge into sequential chunks and iteratively edit the key token in each chunk. The mathematical formulation of the task can be found in Section 3.

Compared with previous studies (Deng et al., 2025; Jiang et al., 2025) that focus on algorithm design and evaluation for unstructured knowledge editing, our work addresses a key gap: the lack of mechanisms to understand target knowledge, which limits performance on hard samples. Instead, we systematically leverage the TANGLESCORE to analyze the editability across diverse instances and propose PIPE to improve model understanding by adapting to instance-specific characteristics.

## 3 PRELIMINARY

Knowledge editing refers to adjusting the knowledge embedded in language models without full retraining. It is widely used to align model outputs with specific tasks or objectives, refining complex learned information such as logical reasoning, spatial understanding, and numerical facts. We study editing knowledge in the form of $(x_e, y_e)$, where a language model $f_\theta \in \mathcal{F}$ is defined as a function $f_\theta : \mathcal{X} \to \mathcal{Y}$ mapping input $x$ to output $y$. When $f_\theta(x_e) \neq y_e$, the goal is to update parameters $\theta \in \Theta$ to obtain an edited model $f_{\theta'}$ such that $f_{\theta'}(x_e) = y_e$. Unlike structured triples, unstructured knowledge involves extended segments of free-form, information-rich text. This makes editing more challenging, as models must comprehend and revise longer, informative content.

Edits often influence a broader range of inputs semantically or syntactically linked to the target, known as the *editing scope*. An effective edit modifies model behavior within this scope while preserving general performance outside of it:

$$f_{\theta'}(x_e) = \begin{cases} y_e & \text{if } x_e \in I(x_e, y_e), \\ f_\theta(x_e) & \text{if } x_e \in O(x_e, y_e). \end{cases}$$

The *in-scope* $I(x_e, y_e)$ includes $x_e$ and its equivalence neighborhood $N(x_e, y_e)$, covering related input/output pairs. The *out-of-scope* $O(x_e, y_e)$ includes inputs unrelated to the edit. To evaluate editing effectiveness in unstructured settings, prior work (Deng et al., 2025; Jiang et al., 2025) focuses on efficacy and generalization, emphasizing both semantic and lexical alignment between edited outputs and target knowledge.

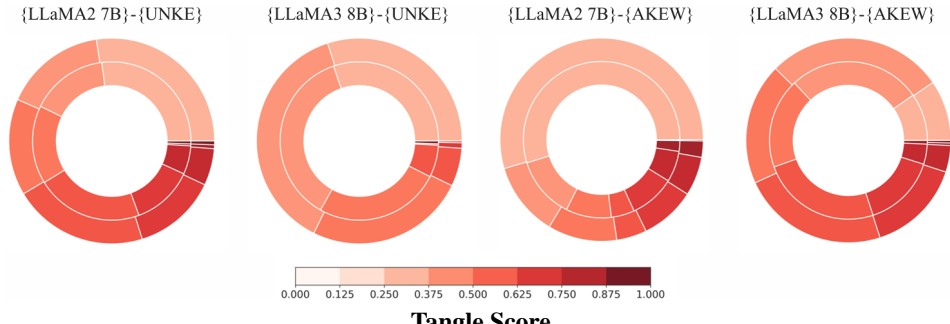

**Figure 2:** Distribution of edited instances' TANGLESCORE using the UNKE method on the UNKEBench and AKEW. Inner rings represent pre-edit distributions; outer rings represent post-edit distributions. Darker colors indicate greater difficulty.

# 4 EVALUATING AND QUANTIFYING SAMPLE-SPECIFIC EDITING DIFFICULTY

Although existing work has made progress in unstructured knowledge editing, it still struggles with certain samples that we identify as particularly difficult to edit (Deng et al., 2025). In this section, we introduce a metric to quantify the ease of editing individual samples, aiming to better understand the factors that influence editing success. Statistical analyzes reveal a strong correlation between the decline in editing performance and the increasing difficulty of these samples, suggesting that existing methods are less effective when handling more challenging cases.

## 4.1 TANGLESCORE

When applying existing unstructured knowledge editing methods, we observed that some samples were successfully edited while others failed. In the failed cases, the model's output remained strongly influenced by the original knowledge, hindering effective updates. These patterns suggest that, for a given LLM, certain samples are inherently more difficult to edit. A core question remains overlooked by current research: *whether there exists a metric that quantifies the resistance of a sample precisely?*

**Internal Representation Shift Analysis** To better understand why certain knowledge edits are more difficult for the model to internalize, we sought to capture the degree of resistance exhibited by each sample during the editing process. Intuitively, when new knowledge is injected, both the model's internal representations and its output responses may change to varying degrees. We aim to quantify this resistance by examining both types of changes. We first considered how the model's *internal hidden states* are affected by the editing process. If the internal representations shift significantly after injecting new knowledge, this may indicate a higher resistance to editing. To measure this, we constructed two input sequences by concatenating the original and novel knowledge with tailored prompts, respectively. The specific construction details can be found in Appendix A. The model's hidden-layer representation, denoted as $r_{\text{old}}$ and $r_{\text{new}}$, were extracted accordingly. Considering the simplicity and effectiveness of cosine similarity in comparing semantic vectors, we computed the semantic distance between their average-pooled representations:

$$D_{\text{semantic}} = 1 - \frac{r_{\text{old}} \cdot r_{\text{new}}}{\|r_{\text{old}}\|\|r_{\text{new}}\|}, \tag{1}$$

A larger value indicates a more significant internal representation shift, thereby implying a stronger resistance to the incorporation and integration of newly injected knowledge.

**Semantic Gap in Output Responses** We also considered the *semantic gap* between the model's answers before and after editing. To evaluate this gap meaningfully, we require a metric that operates in the embedding space, captures global semantic relationships, and remains robust to surface-level differences such as paraphrasing. Unlike KL divergence, which is sensitive to token-level mismatches and often fails to reflect broader semantic alignment, a soft optimal transport approach provides a more holistic comparison by aligning token embeddings across answers. Considering these advantages, we adopt the *Sinkhorn distance* (Watanabe & Isobe, 2024), a differentiable and stable optimal transport metric that preserves contextual meaning while capturing global semantic

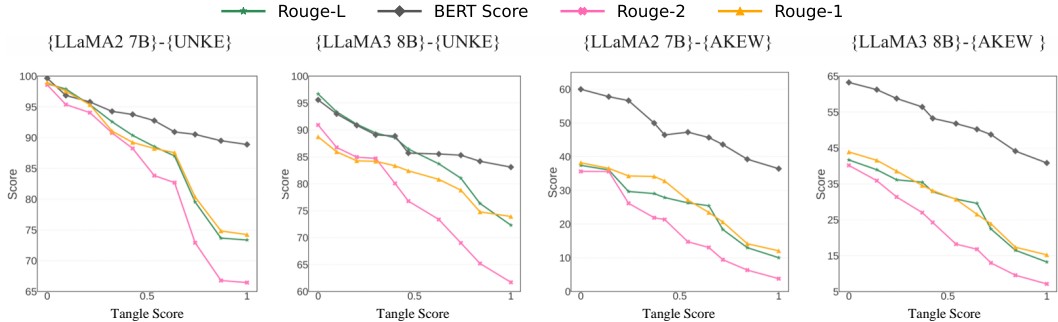

Figure 3: Generalization score variations of ROUGE-1, ROUGE-2, ROUGE-L, and BERT-Score at different levels of TANGLESCORE, using the UNKE method on the UNKEBench and AKEW with the LLaMA3-8B-Instruct and LLaMA2-7B-Chat.

structure. It enables a principled comparison of the two answer distributions in embedding space, aligning them in a semantically meaningful manner.

Taking both components into account, the internal shift and the semantic gap between the origin knowledge and the target answer, we define the TANGLESCORE as a unified metric to quantify how resistant each individual knowledge instance is to editing:

$$\text{TANGLESCORE}(M) = \frac{D_{\text{semantic}}}{\text{Sinkhorn}(\text{Ans}_{\text{old}}, \text{Ans}_{\text{new}})}. \tag{2}$$

**Discussion**    TANGLESCORE characterizes the transition rate of a specific model's propensity to shift from old to new responses by considering two aspects: the change in the model's response behavior when shifting from old to new answers and the distance between the original and updated answers themselves, thereby effectively capturing the degree of entanglement between the to-be-edited knowledge and the model's prior knowledge. This ratio-based formulation reflects the rate of change in the model's responses when the underlying knowledge is modified. To further validate the necessity of this formulation, we conducted more detailed ablation studies, as presented in Appendix D.1. Intuitively, a higher TANGLESCORE value indicates that the model exhibits greater inertia during the editing process, suggesting that such knowledge instances are inherently more difficult to edit. Building on this, we hypothesize that TANGLESCORE influences the generalization of knowledge editing. In section 4.2, we further validate this hypothesis through analysis while revealing several beneficial properties of TANGLESCORE.

## 4.2    CORRELATION OF TANGLESCORE WITH EDITING PERFORMANCE

**TANGLESCORE Reflects Intrinsic Editing Difficulty**    We analyze the behavior of TANGLESCORE and find that it is solely determined by the knowledge to be edited. As shown in Figure 2, we apply the UNKE to models LLaMA3-8B-Instruct (Dubey et al., 2024) and LLaMA2-7B-Chat (Touvron et al., 2023) on UNKEBench (Deng et al., 2025) and AKEW (Wu et al., 2024). Although TANGLESCORE is related to the model's internal representations, which show significant changes before and after editing, we surprisingly observe that the distribution of the TANGLESCORE remains largely unchanged before and after editing. This result suggests that, for different models, the editing difficulty of each knowledge sample is intrinsic and does not vary significantly with changes in the state of the model. Readers can refer to Appendix B.1 for more results of using AlphaEdit (Fang et al., 2025) and RECT (Gu et al., 2024), and to Appendix B.3 for a more rigorous discussion.

**Generalization Declines with Increasing TANGLESCORE**    Given the nature of TANGLESCORE, we hypothesize that editing performance decreases as the TANGLESCORE becomes larger, meaning the knowledge sample is more difficult to edit. To validate our hypothesis, we conducted experiments on LLaMA3-8B-Instruct and LLaMA2-7B-Chat, selecting UNKEBench samples spanning low-to-high TANGLESCORE. To fully assess the generalizability of the model responses after editing with the UNKE method in knowledge samples with varying TANGLESCORE, we comprehensively measured ROUGE scores (Lin, 2004) to evaluate the lexical similarity and BERT scores (Zhang et al., 2020) to assess their semantic similarity between rewritten questions and the standard responses.

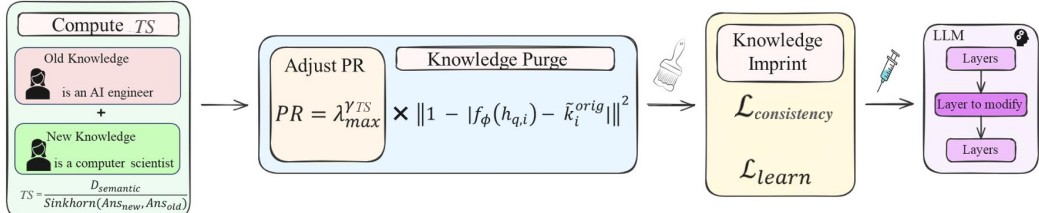

Figure 4: Proposed method: PIPE. We first compute the TANGLESCORE for the knowledge to be edited, then determine the appropriate purge rate and apply it through a knowledge purge function to help the model discard the old information. Subsequently, a knowledge imprint function is used to incorporate the new knowledge, completing the editing process for that specific piece of knowledge.

As shown in Figure 3, ROUGE-Score and BERT-Score all decline monotonically with rising TANGLESCORE values. This indicates that as the difficulty of editing knowledge samples increases, the original UNKE method performs progressively worse in terms of generalization. The decline in both lexical and semantic similarity scores suggests that the model struggles to apply the edited knowledge to alternative phrasings or reworded questions. Readers can refer to Appendix C.1 for more experimental results.

**Discussion**   We argue that the original editing approach does not enable the model to genuinely internalize or comprehend the knowledge being introduced. Instead, it largely forces the model to associate the knowledge with a fixed response pattern, leading to the production of the target answer in a narrow, memorized form. Consequently, when the same piece of knowledge is probed through differently worded or paraphrased prompts, the model frequently fails to generalize and respond appropriately. This limitation suggests that the UNKE method struggles to support edits that are robust, flexible, and transferable across diverse query formulations, thereby raising concerns about its effectiveness in real-world applications where knowledge is naturally expressed in varied ways.

### 4.3 APPLYING TANGLESCORE TO STRUCTURED KNOWLEDGE EDITING

Section 4.2 showed that the TANGLESCORE is an effective measure of the difficulty of unstructured knowledge editing and can help guide the editing process.  This raises the question: *can* TANGLESCORE *also benefit structured knowledge editing?* Accordingly, we apply the UNKE method to LLaMA3-8B-Instruct (Dubey et al., 2024) and LLaMA2-7B-Chat (Touvron et al., 2023) on a structured knowledge editing dataset KEBench (Wei et al., 2024). Experimental results show that the distribution of TANGLESCORE remains largely unchanged before and after editing. This aligns with our observations in unstructured knowledge editing and suggests that structured knowledge also exhibits inherent differences in editing difficulty. Furthermore, we observe that in structured knowledge, the model's generalization performance also exhibits a similar pattern. Readers can find more details in Appendix B.2, Appendix B.3 and Appendix C.2.

## 5   PIPE: PURGE-IMPRINT PATCH EDITING

In Section 4, we propose TANGLESCORE as a measure of the ease of editing a knowledge sample and demonstrate that TANGLESCORE is an intrinsic property of the sample for a given model. Furthermore, our analysis reveals that existing editing methods exhibit poor generalization on more difficult samples, highlighting a key limitation in their effectiveness. As shown in Figure 4, we propose a method called PIPE. This approach enables the model to better understand the knowledge to be edited by first making it purge the old knowledge before learning the new, thereby improving generalization when editing more difficult knowledge samples.

### 5.1   PURGE RATE DETERMINED BY TANGLESCORE

Varying editing difficulties among knowledge samples influence model retention differently, suggesting uniform purge strategies are suboptimal. Since TANGLESCORE quantifies the difficulty of modifying a knowledge sample, we leverage TANGLESCORE to adaptively modulate the purge rate.

Specifically, higher TANGLESCORE implies hard-to-edit knowledge, thus requiring a higher purge rate for effective removal, while lower TANGLESCORE denotes easier samples, where a lower rate avoids overfitting. Accordingly, our final purge mechanism is defined as:

$$PR = \lambda_{max}^{\gamma^{\text{TANGLESCORE}}}, \tag{3}$$

where $\gamma = \log \lambda_{max} / \log \lambda_{min}$ is the scaling factor, and $\lambda_{max}$ and $\lambda_{min}$ represent the maximum and minimum values, respectively. We determined their preset values through preliminary exploratory experiments, and set $\lambda_{max} = 0.001$ and $\lambda_{min} = 0.0001$ in practice. This adaptive purge mechanism aims to mitigate the influence of strongly retained original content, especially when it is difficult to override. To further demonstrate the effectiveness and validity of our approach, we compare it against a baseline with a fixed purge rate. Additional details can be found in Appendix D.2.

## 5.2 KNOWLEDGE PURGE FUNCTION

To mitigate the influence of the original knowledge of the model on learning new information, we introduce a knowledge purge function that encourages the model to effectively unlearn prior knowledge. Inspired by previous work (Liu et al., 2025), we adopt a gradient ascent approach to facilitate purging. By inverting the direction of the gradient, the model is encouraged to increase the loss of information that should be forgotten, thus reducing its predictive accuracy on such data. A natural idea might be to reverse the traditional Mean Squared Error (MSE) loss, which is widely used to minimize the squared difference between a model prediction $\hat{y}$ and a target value $y$:

$$\mathcal{L}_{\text{MSE}} = \|\hat{y} - y\|^2. \tag{4}$$

However, directly maximizing this loss via gradient ascent introduces a critical issue: the gradient of MSE with respect to the model output is proportional to the error term, i.e., $\nabla_{\hat{y}} \mathcal{L}_{\text{MSE}} = 2(\hat{y} - y)$. As the prediction diverges from the target, the gradient grows linearly with $\|\hat{y} - y\|$, which can lead to numerical instability or even gradient explosion during training.

To address this, we introduce a gradient-bounded knowledge purge function that penalizes proximity to original knowledge rather than encouraging large deviations. The proposed formulation avoids MSE's instability by ensuring bounded gradients even as predictions diverge. Considering the purge rate discussed in Section 5.1, we define the final form of our purge loss as a clipped and numerically stable function. Specifically, the absolute difference is bounded within $[0, 1]$ to ensure gradient control. The knowledge purge function for the $i$-th knowledge item is defined as:

$$\mathcal{L}_{\text{purge}} = \Sigma_{i=1}^{u} \text{PR} \cdot \left\| Clamp(1 - \left| f_\phi(h_{q,i}) - \tilde{k}_i^{\text{orig}} \right|) \right\|^2, \tag{5}$$

where $f_\phi(h_{q,i})$ is the current key output for the $i$-th query, $h_{q,i}$ denotes the final hidden state of the $i$-th query, and $\tilde{k}_i^{\text{orig}}$ is the original key vector associated with the knowledge to be purged. This formulation provides a strong unlearning signal by assigning high loss values when the model output remains close to the original key vector, while ensuring stability through gradient saturation as the prediction diverges, effectively preventing unbounded gradients during training.

## 5.3 KNOWLEDGE IMPRINT FUNCTION

After purging outdated knowledge, the model must effectively acquire new knowledge. One of the central challenges in knowledge imprint lies in enabling the model to learn new information while maximally preserving its existing capabilities. This is crucial to minimize the degradation of general abilities and enhance the model's stability in sequential editing. To address this issue, we propose a new consistency $\mathcal{L}_{\text{consistency}}$, which aims to balance knowledge imprint and stability retention during the imprinting process. The objective is formulated as follows:

$$\mathcal{L}_{\text{consistency}} = \Sigma_{i=1}^{u} \Sigma_{j=1}^{n-1} [\alpha \cdot \|f_\phi(h_{q,i,j}) - k_{q,i,j}\|^2 + (1 - \alpha) \cdot \|f_\phi(h_{q,i,j}) - f_\theta(h_{q,i,j})\|^2], \tag{6}$$

where $f_\phi$ denotes the edited model and $f_\theta$ denotes the original model. $h_{q,i,j}$ are hidden states from the $i$-th sample at different token positions in the pre-training and query sequences; $k_{q,i,j}$ are the corresponding target key vectors. $u$ denote the numbers of edited samples. A key component of this loss is the weight $\alpha$, which controls the relative emphasis on knowledge imprint versus stability

preservation at each token position. Rather than setting $\alpha$ as static or binary, we design a dynamic masking mechanism based on the TANGLESCORE:

$$\alpha = \sigma(\text{TANGLESCORE}), \tag{7}$$

where $\sigma$ is the sigmoid function. Intuitively, a higher TANGLESCORE (indicating more difficult edits) leads $\alpha$ to approach 1, making the loss prioritize knowledge transfer. Conversely, a lower TANGLESCORE, which indicates easier edits, causes $\alpha$ to approach 0, thereby favoring the preservation of original capabilities. This design enables $\mathcal{L}_{\text{consistency}}$ to adaptively balance modification and retention during training. In addition, to ensure that the edited model correctly generates the new knowledge, we further incorporate an auxiliary loss:

$$\mathcal{L}_{\text{learn}} = \Sigma_{i=1}^{u} \left\| f_\phi(h_{q,i}) - \tilde{k}_i \right\|^2, \tag{8}$$

The final objective for knowledge imprint is defined as the sum of the two components:

$$\mathcal{L}_{\text{imprint}} = \mathcal{L}_{\text{consistency}} + \mathcal{L}_{\text{learn}}. \tag{9}$$

To enable an end-to-end editing process, we jointly optimize the purge and imprint stages. Ultimately, our final optimization goal is:

$$f_\phi = argmin_\phi(\mathcal{L}_{\text{purge}} + \mathcal{L}_{\text{imprint}}). \tag{10}$$

We selected the 7th layer for editing, and the choice of layer is validated through ablation studies in Appendix D.3. Moreover, to further verify the effectiveness of our proposed optimization terms, we also conducted ablation studies on them. Detailed settings are provided in Appendix D.4.

# 6 EXPERIMENTS

## 6.1 EXPERIMENTAL SETUP

**Base LLMs**   We evaluated on four LLMs: LLaMA3-8B-Instruct (Dubey et al., 2024), LLaMA2-7B-Chat, LLaMA2-13B-Chat (Touvron et al., 2023), and Qwen2.5-7B-Instruct (Yang et al., 2024).

**Baseline Editing Methods**   We compared to ROME (Meng et al., 2022), MEMIT (Meng et al., 2023), RECT (Gu et al., 2024), AlphaEdit (Fang et al., 2025), UNKE (Deng et al., 2025), and AnyEdit (Jiang et al., 2025) as baseline editors.

**Datasets**   Experiments were conducted on three datasets: UNKEBench (Deng et al., 2025), AKEW (Wu et al., 2024) and KEBench (Wei et al., 2024).

**Metrics**   We assessed editing success by comparing edited outputs with targets using both original and paraphrased questions. BERTScore (Zhang et al., 2020) was used to measure semantic similarity, and ROUGE (Lin, 2004) to capture lexical similarity. Additionally, we used FactScore (Min et al., 2023) to measure understanding of new knowledge, and MMLU (Hendrycks et al., 2021) to evaluate general capabilities. Readers can refer to Appendix E for more details.

## 6.2 EVALUATION RESULTS ON UNSTRUCTURED KNOWLEDGE EDITING

This section illustrates the editing performance of edited models with LLaMA3-8B-Instruct, LLaMA2-7B-Chat and Qwen2.5-7B-Instruct on the UNKEBench. Due to page limitations, the detailed experimental results of AKEW and LLaMA2-13B-Chat, the comparison of editing performance under different TANGLESCORE distributions, as well as the comparison of time consumption between PIPE and other methods have been included in Appendix F.

**Editing Performance**   Table 1 presents the editing performance of various methods on the unstructured knowledge dataset UNKEBench. Traditional editing methods designed for structured knowledge, such as ROME, MEMIT, RECT, and AlphaEdit, showed significantly lower performance on UNKEBench. These methods struggled with challenges in unstructured knowledge editing, particularly compared to approaches developed for unstructured settings. In contrast, PIPE achieved notable improvements over existing unstructured knowledge editing methods, including UNKE and AnyEdit, with higher generalization performance, suggesting it helps the model better capture and internalize the knowledge to be edited. Our method also demonstrated consistent robustness across sequential and batch editing. Detailed results are provided in Appendix G.

Table 1: Editing performance on UNKE across different methods. The batch size was fixed at 1, with model parameters rebuilt after each edit. Decoding used a temperature setting of 0.001. To ensure fair comparison, edits were applied to the 7th layer of parameters for ROME, RECT, and UNKE. Evaluation results before and after the '/' represent outputs for original and paraphrased questions, respectively. FC refers to Factual Correctness.

| Method | Semantic Similarity | Lexical Similarity | | | FC | General Ability |
|--------|---------------------|--------------------|---|---|----|-----------------|
|        | Bert-Score          | Rouge-1 | Rouge-2 | Rouge-L | FactScore | MMLU |
| **LLaMA3-8B-Instruct** | | | | | | 29.94 |
| ROME | 72.33 / 71.29 | 48.88 / 42.31 | 28.46 / 21.75 | 45.87 / 40.11 | 23.67 | 29.59 |
| RECT | 73.79 / 71.44 | 48.42 / 42.51 | 29.64 / 21.71 | 46.28 / 40.96 | 23.55 | 29.33 |
| MEMIT | 76.79 / 74.46 | 44.91 / 41.82 | 25.38 / 20.04 | 40.01 / 36.57 | 25.34 | 29.44 |
| AlphaEdit | 75.28 / 73.65 | 45.02 / 41.67 | 25.77 / 18.84 | 41.93 / 36.73 | 28.91 | 29.03 |
| UNKE | 97.41 / 90.06 | 98.54 / 78.71 | 98.17 / 70.83 | 97.86 / 77.72 | 41.56 | 29.45 |
| AnyEdit | 98.62 / 91.56 | 97.20 / 78.62 | 96.29 / 72.59 | 95.15 / 79.60 | 48.48 | 28.52 |
| PIPE | **98.64 / 91.71** | **98.48 / 85.42** | **98.89 / 78.03** | **98.44 / 84.07** | **50.91** | 29.51 |
| **LLaMA2-7B-Chat** | | | | | | 29.71 |
| UNKE | 96.56 / 90.73 | 98.51 / 77.98 | 97.84 / 67.24 | 95.34 / 76.11 | 40.99 | 29.07 |
| AnyEdit | 98.37 / 91.60 | 95.69 / 78.55 | 96.04 / 72.17 | 93.59 / 75.68 | 47.82 | 29.66 |
| PIPE | **98.57 / 91.73** | **98.66 / 84.05** | **98.21 / 78.12** | **97.39 / 78.47** | **50.12** | 29.65 |
| **Qwen2.5-7B-Chat** | | | | | | 32.63 |
| UNKE | 96.84 / 89.92 | 92.55 / 72.81 | 90.77 / 69.94 | 93.64 / 73.89 | 40.12 | 31.28 |
| AnyEdit | 97.10 / 90.71 | 93.84 / 75.26 | 92.90 / 73.39 | 95.38 / 77.14 | 40.60 | 30.47 |
| PIPE | **97.42 / 91.76** | **94.71 / 78.63** | **93.88 / 75.40** | **97.05 / 80.18** | **42.47** | 31.78 |

**Knowledge Comprehension and General Abilities Preservation** Our goal is to help the model better understand newly imprinted knowledge when purging outdated information, without sacrificing general abilities. As shown in Table 1, in the multi-hop comprehension task on the edited knowledge, PIPE showed stronger comprehension compared to baselines, suggesting it encourages the model to internalize and reason about new knowledge beyond simple memorization. Compared to the pre-edited model, PIPE preserved general abilities, indicating that selective purging does not degrade broader performance on unseen data. These results highlight PIPE's effectiveness in balancing knowledge integration with robust generalization. For more generated cases, see Appendix H.

## 6.3 EVALUATION RESULTS ON STRUCTURED KNOWLEDGE EDITING

To demonstrate that our approach is also robust for editing traditional structured knowledge, we conducted experiments on the KEBench. As shown in Table 2, PIPE achieves significant improvements on both the Ori-Acc and Para-Acc metrics compared to existing structured and unstructured knowledge editing methods. These results clearly demonstrated that our method can be successfully and robustly applied to both structured and unstructured knowledge editing tasks, offering a more unified and generalizable solution to knowledge editing challenges.

Table 2: Structured editing performance on KEBench. Ori-Acc and Para-Acc represent the accuracy for original and paraphrased questions, respectively. Src-Acc and Tgt-Acc represent the irrelevant knowledge accuracy of subject and object in structured triplets, respectively.

| Method | Ori-Acc | Para-Acc | Src-Acc | Tgt-Acc |
|--------|---------|----------|---------|---------|
| ROME | 76.35 | 67.23 | 95.54 | 74.57 |
| RECT | 76.89 | 68.97 | 95.98 | 75.84 |
| MEMIT | 74.33 | 65.08 | **97.67** | 77.48 |
| AlphaEdit | 75.45 | 66.83 | 97.52 | **78.06** |
| UNKE | 93.59 | 85.34 | 89.28 | 62.56 |
| PIPE | **95.89** | **88.23** | 94.47 | 70.11 |

## 6.4 ANALYSIS OF PIPE

To further illustrate why PIPE achieves better editing performance on harder knowledge, we conducted additional analysis. As shown in Figure 5(a), the distribution of TANGLESCORE remains largely unchanged before and after applying PIPE edits, further confirming that TANGLESCORE is an intrinsic property of each knowledge instance and is not altered by PIPE. In addition, we concatenated the previous answer with the prompt as input to the model, from which we obtain the probability of the model outputting old knowledge. From Figure 5(b) we could find that the probability distribution after editing shifts to the right overall, indicating that PIPE suppresses the model's tendency to output old knowledge more effectively. We concluded that through PIPE's purging mechanism, the model better refrained from relying on old knowledge, thereby improving its understanding of the knowledge to be edited.

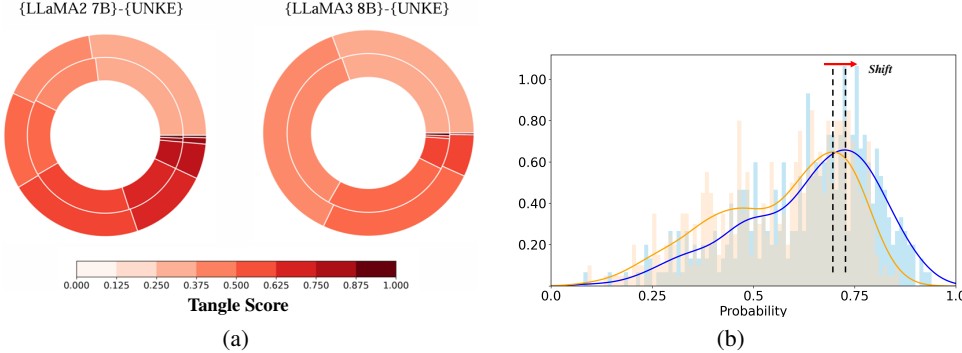

Figure 5: (a) Distribution of TANGLESCORE using PIPE on the UNKEBench. (b) Comparison of the model's tendency to output old knowledge under UNKE and PIPE editing methods, using negative log-probability to make the distribution more interpretable. The curves represent the results of KDE.

## 7 CONCLUSION & LIMITATION

We revisit knowledge editing by examining the gap between accuracy and generalization through the lens of editing difficulty. To quantify this, we propose TANGLESCORE, a metric that strongly correlates with generalization performance post-edit. Leveraging TANGLESCORE, we present PIPE, a framework that adaptively selects editing strategies based on knowledge difficulty. This method consistently improves both editing efficacy and model generalization. Extensive experiments across diverse settings demonstrate the robustness of PIPE. In addition, TANGLESCORE offers a novel lens for understanding editing challenges, emphasizing how editing difficulty shapes post-editing behavior. It serves as a practical tool for analyzing and enhancing editing techniques. Overall, our work advances editing strategies and lays the groundwork for future knowledge editing research.

Despite the effectiveness of PIPE, our work has several limitations. First, while PIPE performs well in controlled settings, it does not yet support continuous editing involving sequential knowledge updates. Second, it is limited to textual editing and lacks support for multimodal integration. Extending it to handle cross-modal updates is a promising direction. Lastly, further validation on larger models and more diverse datasets is needed to better assess PIPE's generality.

### ACKNOWLEDGMENTS

We would like to express gratitude to the anonymous reviewers for kind comments. This work was supported in part by the New Generation Artificial Intelligence-National Science and Technology Major Project (No. 2025ZD0123204).

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

## A CONSTRUCTION OF ORIGINAL AND NOVEL KNOWLEDGE WITH PROMPTS

To obtain the hidden-layer representation $r_{\text{old}}$ and $r_{\text{new}}$, as shown in Table 3, we concatenated the original knowledge and the target knowledge with the corresponding question, respectively, resulting in two input sequences.

Table 3: Construction details of input sequences

| Prompt | "What is John Finlay's occupation and what supports his expertise?" |
|---|---|
| Original Knowledge | "John Finlay is a former CIA officer..." |
| Novel Knowledge | "John Finlay is a police officer with..." |
| Input Old | "Prompt: {*Prompt*}\nAnswer: {*Original Knowledge*}" |
| Input New | "Prompt: {*Prompt*}\nAnswer: {*Novel Knowledge*}" |

## B DISTRIBUTION OF EDITED SAMPLES BY TANGLESCORE

### B.1 DISTRIBUTION OF EDITED SAMPLES BY TANGLESCORE ON UNSTRUCTURED KNOWLEDGE

To further demonstrate that TANGLESCORE depends solely on the model and the knowledge to be edited, rather than the editing method, we apply different editing methods and analyze the distribution of TANGLESCORE across the dataset before and after editing. As shown in Figure 6 and Figure 7, we apply the AlphaEdit (Fang et al., 2025) and RECT (Gu et al., 2024) to LLaMA3-8B-Instruct (Dubey et al., 2024) and LLaMA2-7B-Chat (Touvron et al., 2023) , respectively, conducting experiments on UNKEBench (Deng et al., 2025). The results of this experiment are consistent with our previous findings, further reinforcing the robustness of our conclusions: for different models, the editing difficulty of each knowledge sample is intrinsic and does not vary significantly with changes in the state of the model.

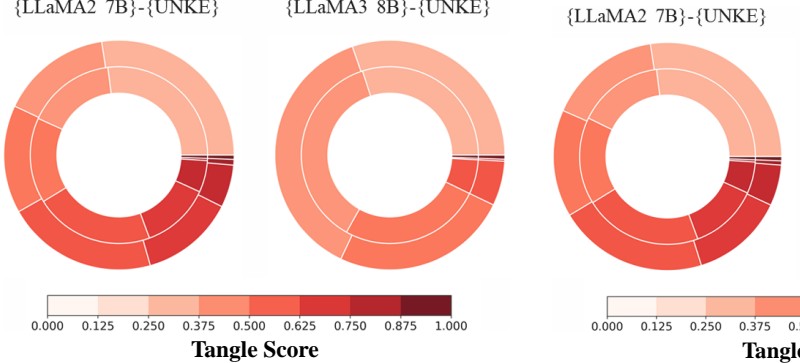

Figure 6: Distribution of edited knowledge samples across TANGLESCORE ranges, shown for LLaMA3-8B-Instruct and LLaMA2-7B-Chat using the AlphaEdit on UNKEBench. Inner rings represent pre-edit distributions; outer rings represent post-edit distributions.

Figure 7: Distribution of edited knowledge samples across TANGLESCORE ranges, shown for LLaMA3-8B-Instruct and LLaMA2-7B-Chat using the RECT on UNKEBench. Inner rings represent pre-edit distributions; outer rings represent post-edit distributions.

### B.2 DISTRIBUTION OF EDITED SAMPLES BY TANGLESCORE ON STRUCTURED KNOWLEDGE

We also conducted the above experiments on structured knowledge. As shown in Figure 8, we applied UNKE (Deng et al., 2025) to LLaMA3-8B-Instruct (Dubey et al., 2024) and LLaMA2-7B-Chat (Touvron et al., 2023), respectively, and conducted experiments on KEBench (Wei et al., 2024). The results on structured knowledge are consistent with those on unstructured knowledge, further reinforcing the robustness of our conclusions.

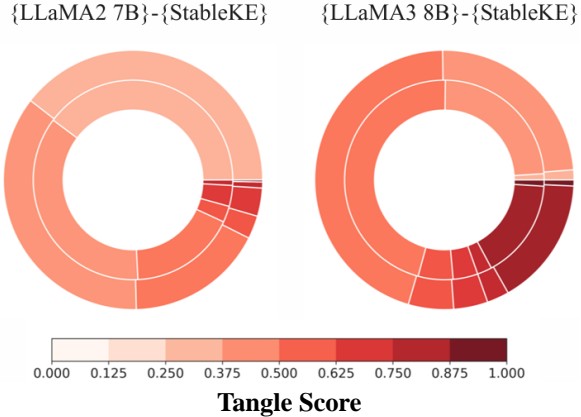

Figure 8: Distribution of edited knowledge samples across TANGLESCORE ranges using the UNKE method on the KEBench. Inner rings represent pre-edit distributions; outer rings represent post-edit distributions. Darker colors indicate greater difficulty.

### B.3 FURTHER ANALYSIS OF EDITED SAMPLE DISTRIBUTION BY TANGLESCORE

To more rigorously and comprehensively support our conclusion that for different models, the editing difficulty of each knowledge sample is intrinsic and does not vary significantly with changes in the state of the model, we further investigate the cross-model correlation of TANGLESCORE on the same samples. Figure 9 and Figure 10 present the TANGLESCORE values of individual knowledge samples before and after editing on the unstructured knowledge and structured knowledge datasets, respectively. The x-axis denotes the pre-edit TANGLESCORE, while the y-axis denotes the post-edit TANGLESCORE. As shown in the figures, for both LLaMA3-8B-Instruct (Dubey et al., 2024) and LLaMA2-7B-Chat (Touvron et al., 2023), the TANGLESCORE values of individual samples remain largely unchanged before and after editing. This further reinforces the robustness of our conclusion.

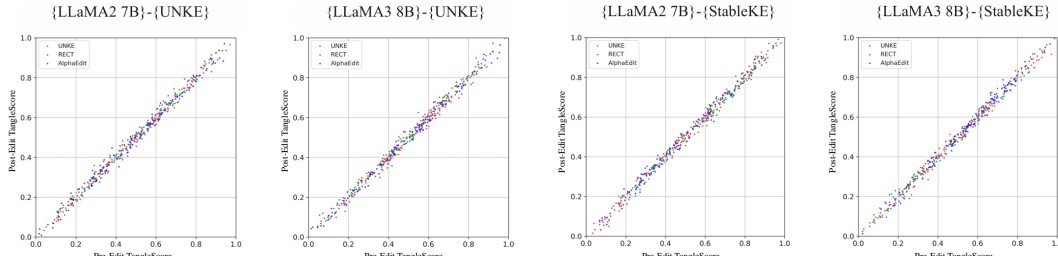

Figure 9: Scatter plot of TANGLESCORE changes before and after single-sample editing. We randomly sampled 200 examples from UNKEBench, using UNKE, RECT, and AlphaEdit.

Figure 10: Scatter plot of TANGLESCORE changes before and after single-sample editing. We randomly sampled 200 examples from StableKE, using UNKE, RECT, and AlphaEdit.

## C APPLYING TANGLESCORE TO KNOWLEDGE EDITING

### C.1 APPLYING TANGLESCORE TO UNSTRUCTURED KNOWLEDGE EDITING

To demonstrate that TANGLESCORE can effectively measure the difficulty of unstructured knowledge editing and thereby help guide more effective edits on challenging samples, we conducted additional experiments on the UNKEBench (Deng et al., 2025). Specifically, we applied the RECT (Gu et al., 2024) and the AlphaEdit (Fang et al., 2025) to LLaMA3-8B-Instruct (Dubey et al., 2024). Figure 11 illustrates how the editing generalization score varies with different TANGLESCORE magnitudes.

## C.2 Applying TangleScore to Structured Knowledge Editing

To demonstrate that TangleScore can effectively measure the difficulty of structured knowledge editing and thereby help guide more effective edits on challenging samples, we conducted experiments on the structured knowledge dataset KEBench (Wei et al., 2024). Specifically, we applied the UNKE (Deng et al., 2025) to LLaMA3-8B-Instruct (Dubey et al., 2024) and LLaMA2-7B-Chat (Touvron et al., 2023). Figure 12 illustrates how the editing generalization score varies with different TangleScore magnitudes.

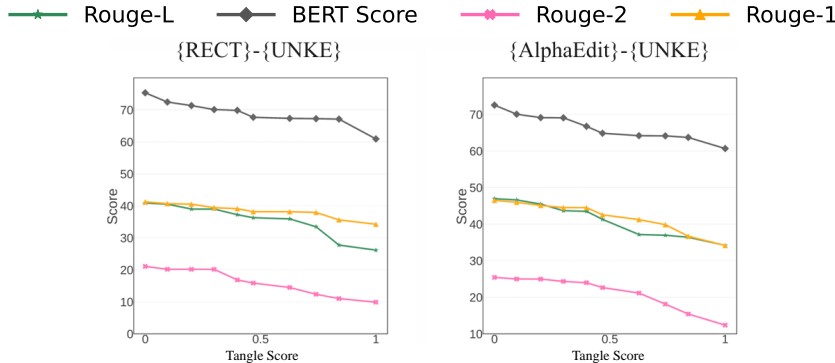

Figure 11: Generalization score variations of ROUGE-1, ROUGE-2, ROUGE-L, and BERT-Score at different levels of TangleScore, using the RECT and AlphaEdit method on the UNKEBench with the LLaMA3-8B-Instruct.

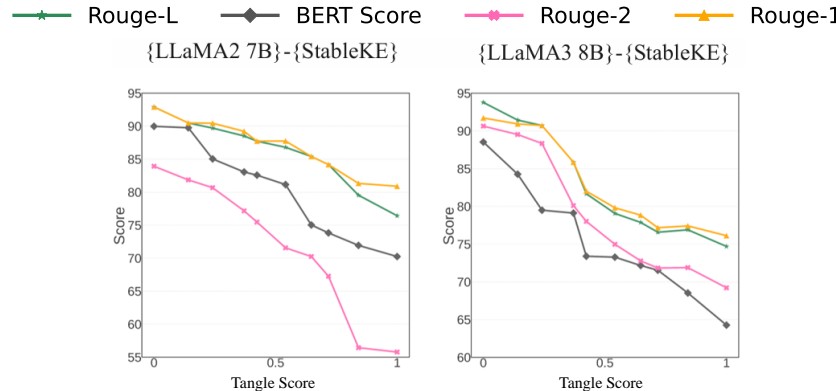

Figure 12: Generalization score variations of ROUGE-1, ROUGE-2, ROUGE-L, and BERT-Score at different levels of TangleScore, using the UNKE method on the KEBench with the LLaMA3-8B-Instruct and LLaMA2-7B-Chat.

## D Ablation Studies

### D.1 Impact of TangleScore Components

TangleScore consists of two components: the numerator captures changes in the model's internal responses, while the denominator reflects the semantic distance between the old and new knowledge. The ratio indicates how strongly the model's responses will change per unit of knowledge modification. In this sense, the ratio-based formulation is necessary. To further support this claim, we conducted an ablation study in which only the numerator or only the denominator was used to guide the purge rate during editing, with results presented in Table 4. It can be observed that using either the numerator or the denominator alone leads to a substantial drop in editing performance. This is because, under these settings, the editing process either considers only the model's responses or only the difference

Table 4: Ablation study on TangleScore components using only the numerator or only the denominator. Experiments are conducted on UNKEBench using LLaMA3-8B-Instruct.

| Method | Semantic Similarity | Lexical Similarity | | | FC | General Ability |
|---|---|---|---|---|---|---|
| | Bert-Score | Rouge-1 | Rouge-2 | Rouge-L | FactScore | MMLU |
| Numerator Only | 71.80 / 55.48 | 73.59 / 62.81 | 67.68 / 47.28 | 59.59 / 53.79 | 32.79 | 28.48 |
| Denominator Only | 64.61 / 59.12 | 76.66 / 52.74 | 63.66 / 54.70 | 71.86 / 59.60 | 36.55 | 27.74 |
| Ratio Form | **98.64 / 91.71** | **98.48 / 85.42** | **98.89 / 78.03** | **98.44 / 84.07** | **50.91** | 29.51 |

Table 5: Comparison of PIPE's performance under different purge rate strategies: "Fixed PR (max)" uses a fixed rate of 0.001, "Fixed PR (min)" uses 0.0001, and "Dynamic PR" adapts the rate based on sample difficulty. Evaluation results before and after the '/' represent outputs for original and paraphrased questions, respectively. FC refers to Factual Correctness.

| Method | Semantic Similarity | Lexical Similarity | | | FC | General Ability |
|---|---|---|---|---|---|---|
| | Bert-Score | Rouge-1 | Rouge-2 | Rouge-L | FactScore | MMLU |
| Fixed PR(max) | 98.03 / 90.26 | 98.34 / 80.00 | 98.31 / 74.79 | 97.75 / 80.37 | 48.21 | 28.71 |
| Fixed PR(min) | 97.50 / 88.14 | 96.85 / 78.39 | 97.57 / 71.13 | 95.99 / 78.62 | 43.85 | 29.22 |
| Dynamic PR | **98.64 / 91.71** | **98.48 / 85.42** | **98.89 / 78.03** | **98.44 / 84.07** | **50.91** | 29.51 |

between old and new knowledge, causing TANGLESCORE to lose its original meaning and fail to properly guide the purge rate.

## D.2 THE EFFECTIVENESS OF PR

To further demonstrate the effectiveness of dynamically adjusting the purge rate based on TANGLESCORE, we compared it with approaches that use a fixed purge rate. As shown in Table 5, whether a high or low fixed rate is applied, the editing performance consistently underperforms compared to the dynamic approach, especially in the generalizability metric. This result suggests that adapting the purge strategy to the difficulty level of each knowledge instance leads to more accurate and robust editing outcomes.

## D.3 IMPACT OF THE SELECTED LAYER ON MODEL PERFORMANCE

In Section 5, we designate the 7th layer as the editing layer. Previous studies have shown that relatively shallow layers are more likely to preserve model stability after editing, whereas deeper layers contain more entangled target information in the key vector, making it difficult to modify without impairing the model's original abilities (Deng et al., 2025). To examine the impact of this design choice on model performance, we conducted a analytical study, and the results are presented in Table 6. The findings show that editing the 7th layer achieves the most stable post-editing performance while maintaining the model's ability to handle unstructured knowledge. In addition, no special processing is applied to the embeddings of old and new answers. We directly use the model's tokenizer to obtain standard token embeddings to ensure full consistency with the normal inference procedure.

Table 6: Ablation study of computing TangleScore at different layers and performing edits. Experiments are conducted on UNKEBench using LLaMA3-8B-Instruct.

| Layer | Semantic Similarity | Lexical Similarity | | | FC | General Ability |
|---|---|---|---|---|---|---|
| | Bert-Score | Rouge-1 | Rouge-2 | Rouge-L | FactScore | MMLU |
| 4 | 94.38 / 87.47 | 93.96 / 82.37 | 94.55 / 75.28 | 94.07 / 80.23 | 48.27 | 28.19 |
| 5 | 96.51 / 89.32 | 96.17 / 83.86 | 96.74 / 76.94 | 96.28 / 82.61 | 49.12 | 28.87 |
| 6 | 97.28 / 90.05 | 97.11 / 84.62 | 97.43 / 77.22 | 97.09 / 83.01 | 50.83 | 29.18 |
| 7 | **98.64 / 91.71** | **98.48 / 85.42** | **98.89 / 78.03** | **98.44 / 84.07** | **50.91** | 29.51 |
| 8 | 97.05 / 89.14 | 96.82 / 83.12 | 97.11 / 76.65 | 96.78 / 82.34 | 49.47 | 29.07 |
| 9 | 96.12 / 88.09 | 95.94 / 82.01 | 96.41 / 75.86 | 96.10 / 81.79 | 48.69 | 28.98 |
| 10 | 95.21 / 87.03 | 95.03 / 80.56 | 95.62 / 74.49 | 95.20 / 80.12 | 48.11 | 28.63 |

Table 7: Ablation study of the individual loss functions. Experiments are conducted on UNKEBench using LLaMA3-8B-Instruct.

| Method | Semantic Similarity | Lexical Similarity | | | FC | General Ability |
|---|---|---|---|---|---|---|
| | Bert-Score | Rouge-1 | Rouge-2 | Rouge-L | FactScore | MMLU |
| w/o $\mathcal{L}_{\text{purge}}$ | 96.33 / 90.19 | 92.27 / 80.26 | 91.44 / 71.76 | 91.92 / 80.51 | 47.15 | 28.73 |
| w/o $\mathcal{L}_{\text{consistency}}$ | 37.86 / 35.12 | 15.64 / 17.93 | 23.56 / 27.77 | 39.84 / 42.63 | 26.61 | 28.91 |
| w/o $\mathcal{L}_{\text{purge}}$ and $\mathcal{L}_{\text{consistency}}$ | 12.07 / 13.88 | 6.54 / 6.75 | 14.96 / 15.13 | 18.59 / 22.17 | 14.67 | 26.35 |

## D.4 IMPACT OF DIFFERENT LOSS FUNCTIONS

To further illustrate the roles of the functions we propose, we conducted an ablation study. As shown in Table 7, removing the $\mathcal{L}_{\text{purge}}$ leads to a noticeable drop in the generalization of edits, highlighting the importance of this loss function. Moreover, when the $\mathcal{L}_{\text{consistency}}$ is removed, both the accuracy and generalization of the edits decrease significantly, indicating that leveraging contextual information is crucial during the process of imprinting new knowledge.

## E  EXPERIMENTAL SETUP

### E.1  BASELINE METHODS

In our experiments, six popular knowledge editing methods were selected as baselines, including:

- **ROME** (Meng et al., 2022)[1]: it first localized the factual knowledge at a specific layer in the transformer MLP modules, and then updated the knowledge by directly writing new key-value pairs in the MLP module.

- **MEMIT** (Meng et al., 2023)[2]: it extended ROME to edit a large set of facts and updated a sequence of MLP layers to update knowledge.

- **RECT** (Gu et al., 2024)[3]: it is designed to reduce the unintended side effects of knowledge editing on the general capabilities of large language models (LLMs). While editing can enhance factual accuracy, it often harms performance on tasks such as reasoning and question answering. RECT tackles this problem by regularizing weight updates during the editing process, thereby limiting excessive changes that may cause overfitting. As a result, RECT achieves strong editing performance while preserving the model's overall abilities.

- **AlphaEdit** (Fang et al., 2025)[4]: It mitigates catastrophic forgetting during knowledge editing by preserving previously learned knowledge, and extends this capability to a lifelong editing setting through a null-space projection strategy. This approach ensures that new edits do not interfere with prior information by projecting updates into directions that minimally affect existing representations, thereby maintaining model stability across a long sequence of edits.

- **UNKE** (Deng et al., 2025)[5]: it enhances knowledge editing through innovations at both the layer and token levels. Specifically, it replaces traditional local key-value storage with a non-local block structure to better capture and integrate attention-based knowledge. On the token level, it shifts from term-driven to cause-driven optimization, directly editing the final token to improve edit precision while preserving contextual coherence.

- **AnyEdit** (Jiang et al., 2025)[6]: it introduces an auto-regressive editing paradigm that supports long-form and diverse-formatted knowledge updates. By chunking outputs and editing one token per chunk, it avoids interference between edits and ensures coherence. Grounded in mutual information theory, AnyEdit adapts to various formats (e.g., code, math) and overcomes the single-token editing barrier, offering strong scalability and generalization.

---

[1]https://github.com/kmeng01/rome

[2]https://github.com/kmeng01/memit

[3]https://github.com/JasonForJoy/Model-Editing-Hurt

[4]https://github.com/ jianghoucheng/AlphaEdit

[5]https://github.com/TrustedLLM/UNKE

[6]https://github.com/ jianghoucheng/AnyEdit

### E.2 Editing Datasets

In our experiments, UNKEBench, AKEW benchmark and KEBench were adopted.

- **UNKEBench** (Deng et al., 2025): it provides a dataset of 1,000 counterfactual unstructured texts, where knowledge is expressed in a relatively long and free-form manner, extending beyond simple knowledge triplets or linear factual chains. These samples are derived from ConflictQA (Xie et al., 2023), a benchmark designed to differentiate between knowledge stored in LLMs' parameter memory and anti-memory. This setup is essential to avoid the blending of pretrained knowledge with information introduced during editing. It also tackles the critical challenge of verifying whether the model acquires specific knowledge through pretraining or through post-hoc editing, thereby maintaining a clear separation.

- **AKEW benchmark** (Wu et al., 2024): it considers three types of knowledge: (1) *Structured Facts*: Each structured fact is an individual triplet intended for editing, drawn from existing datasets or knowledge graphs. (2) *Unstructured Facts*: Knowledge is provided in natural language form. To ensure fair comparison, each unstructured instance conveys the same factual update as its structured counterpart. Unstructured facts are more linguistically complex, often embedding implicit information that goes beyond explicit triplets. (3) *Extracted Triplets*: These are automatically derived from unstructured texts to examine whether such representations can assist editing methods in managing free-text knowledge.

- **KEBench** (Wei et al., 2024): It introduces a comprehensive benchmark specifically designed to evaluate the stability of structured knowledge editing. The dataset includes 1,000 editing triplets and 2,798 multi-hop questions, organized in a tree-structured format to reflect realistic, hierarchical reasoning scenarios. KEBench captures four critical dimensions of stability: edited knowledge consistency, multi-hop reasoning integrity, preservation of unrelated knowledge, and maintenance of general model capabilities. Unlike previous benchmarks, it emphasizes both large-scale editing (batch and sequential) and complex reasoning paths, offering a robust and scalable framework for assessing structured knowledge editing methods in large language models.

### E.3 Evaluation Metrics

We evaluated the performance of our unstructured editing across four dimensions: word-level overlap, semantic similarity, factual correctness, and general abilities, primarily following established benchmarks for unstructured knowledge editing.

- **Lexical Similarity** metrics, including ROUGE scores (ROUGE-1, ROUGE-2, and ROUGE-L) (Lin, 2004), evaluate the degree of lexical and n-gram overlap between model-generated responses and target answers, for both original and paraphrased inputs. These metrics are essential for assessing the surface-level accuracy of the edited content.

- **Semantic Similarity**. Since metrics based solely on word overlap fail to capture deeper understanding, we employ a semantic evaluation approach. Specifically, we use an embedding-based encoder (all-MiniLM-L6-v2[7]) to measure how well the model comprehends the intended meaning, providing a more comprehensive evaluation that moves beyond surface-level matching.

- **Factual Correctness**. To determine whether the edited model accurately grasps unstructured knowledge, we adopt FactScore (Min et al., 2023) to measure its performance on sub-questions and corresponding answers. This evaluation offers a more fine-grained perspective and is conceptually aligned with multi-hop accuracy used in certain structured knowledge editing tasks.

- **General Ability**. To evaluate overall capability, we measure the MMLU score of the edited model. Following Hendrycks et al. (2021), we first compute the average MMLU score across five samples for each unstructured instance, and then derive an overall average across all samples.

---

[7]https://huggingface.co/sentence-transformers/all-MiniLM-L6-v2

### E.4 EXPERIMENTS COMPUTE RESOURCES

We conducted our experiments on a single NVIDIA A800 80GB GPU. It occupies about 40+GB memory for LLaMA2-7B-Chat and about 50+GB memory for LLaMA3-8B-Instruct. For the time costs of one edit, our PIPE is about 1 minute with LLaMA2-7B-Chat. The time cost of LLaMA3-8B-Instruct is 45% higher than LLaMA2-7B-Chat.

Table 8: Editing performance on AKEW (Counterfact) across different methods. The batch size was fixed at 1, with model parameters rebuilt after each edit. Decoding used a temperature setting of 0.001. To ensure fair comparison, edits were applied to the 7th layer of parameters for ROME, RECT, and UNKE. Evaluation results before and after the '/' represent outputs for original and paraphrased questions, respectively.

| Method | Semantic Similarity | Lexical Similarity | | | General Ability |
|--------|---------------------|--------------------|--|--|-----------------|
| | Bert-Score | Rouge-1 | Rouge-2 | Rouge-L | MMLU |
| **LLaMA3-8B-Instruct** | | | | | 29.94 |
| ROME | 72.87 / 41.36 | 27.48 / 14.26 | 12.15 / 11.29 | 30.40 / 14.95 | 29.60 |
| RECT | 73.94 / 41.91 | 29.88 / 15.63 | 13.85 / 13.52 | 30.42 / 15.16 | 29.67 |
| MEMIT | 76.59 / 46.12 | 32.43 / 18.72 | 16.34 / 10.64 | 33.84 / 17.55 | 29.43 |
| AlphaEdit | 76.11 / 46.78 | 26.21 / 15.53 | 13.00 / 9.12 | 27.44 / 14.53 | 29.44 |
| UNKE | 97.94 / 56.73 | 94.96 / 34.35 | 93.33 / 17.88 | 93.43 / 32.69 | 29.76 |
| AnyEdit | 98.37 / 61.22 | 95.24 / 44.08 | 93.17 / 27.72 | 94.71 / 45.81 | 29.52 |
| PIPE | **99.15 / 65.72** | **95.67 / 46.92** | **94.51 / 30.11** | **95.61 / 48.38** | 29.85 |
| **LLaMA2-7B-Chat** | | | | | 29.71 |
| UNKE | 96.81 / 55.93 | 86.66 / 30.90 | 84.98 / 11.54 | 86.35 / 30.85 | 29.16 |
| AnyEdit | 96.18 / 60.09 | 88.71 / 41.05 | 87.35 / 23.88 | 90.46 / 35.91 | 29.34 |
| PIPE | **98.04 / 63.37** | **89.29 / 44.02** | **87.75 / 26.97** | **92.05 / 40.10** | 29.62 |
| **Qwen2.5-7B-Chat** | | | | | 32.63 |
| UNKE | 95.66 / 54.34 | 93.32 / 33.59 | 90.57 / 15.81 | 91.22 / 29.50 | 31.88 |
| AnyEdit | 96.80 / 59.92 | 95.11 / 42.62 | 91.01 / 26.35 | 92.90 / 43.04 | 31.53 |
| PIPE | **97.91 / 63.52** | **97.38 / 44.11** | **92.81 / 27.59** | **94.09 / 45.76** | 32.12 |

## F EXPERIMENTAL RESULTS

### F.1 THE RESULTS OF AKEW

Tables 8 and 9 present the editing performance of various methods on AKEW (Counterfact) and AKEW (MQUAKE), respectively. The results align with the findings from previous experiments. Compared to other baselines, our method, PIPE, demonstrates improved editing performance, with particularly notable gains in the generalizability metric.

### F.2 THE RESULTS OF LLAMA2-13B-CHAT

Tables 10 present the editing performance of various methods on LLaMA2-13B-Chat (Touvron et al., 2023). The experimental results demonstrate that our method outperforms previous baselines even when applied to relatively large-scale models, further highlighting its robustness and effectiveness.

### F.3 COMPARISON OF EDITING PERFORMANCE ACROSS DIFFERENT TANGLESCORE DISTRIBUTIONS

In our previous analysis, we noted that for samples with low TANGLESCORE, existing editing methods are generally able to handle them well. Our PIPE method mainly targets difficult samples with high TANGLESCORE. Experimental results also demonstrate that PIPE achieves a significant improvement in editing performance, which is clearly due to its greater advantage on high TANGLESCORE samples. To illustrate this in more detail, we further divided the knowledge to be edited into four groups according to TANGLESCORE and compared the editing performance of different methods on each group of samples. The results are reported in Table 11. It should be noted that the TANGLESCORE ranges in the table are normalized using the min-max method. The results show that for samples with low TANGLESCORE, most editing methods achieve good performance, while PIPE still obtains the

Table 9: Editing performance on AKEW (MQUAKE) across different methods. The batch size was fixed at 1, with model parameters rebuilt after each edit. Decoding used a temperature setting of 0.001. To ensure fair comparison, edits were applied to the 7th layer of parameters for ROME, RECT, and UNKE. Evaluation results represent outputs for original questions.

| Method | Semantic Similarity | Lexical Similarity | | | General Ability |
| --- | --- | --- | --- | --- | --- |
| | Bert-Score | Rouge-1 | Rouge-2 | Rouge-L | MMLU |
| **LLaMA3-8B-Instruct** | | | | | 29.94 |
| ROME | 71.80 | 25.77 | 11.28 | 26.78 | 29.32 |
| RECT | 70.67 | 24.96 | 13.54 | 27.08 | 29.05 |
| MEMIT | 72.98 | 26.73 | 15.40 | 29.02 | 29.31 |
| AlphaEdit | 72.76 | 27.82 | 14.97 | 29.34 | 29.19 |
| UNKE | 97.43 | 93.99 | 91.51 | 92.82 | 29.04 |
| AnyEdit | 96.91 | 94.18 | 92.56 | 93.94 | 29.22 |
| PIPE | **97.91** | **94.31** | **94.56** | **95.25** | 29.44 |
| **LLaMA2-7B-Chat** | | | | | 29.71 |
| UNKE | 95.06 | 84.18 | 82.55 | 85.33 | 28.73 |
| AnyEdit | 95.49 | 85.00 | 84.24 | 86.42 | 29.42 |
| PIPE | **96.41** | **85.17** | **85.22** | **87.63** | 29.68 |
| **Qwen2.5-7B-Chat** | | | | | 32.63 |
| UNKE | 95.31 | 91.57 | 88.66 | 92.02 | 31.86 |
| AnyEdit | 95.88 | 92.65 | 89.01 | 92.57 | 31.16 |
| PIPE | **96.72** | **95.92** | **90.74** | **92.58** | 32.03 |

Table 10: Editing performance on UNKE across different methods. The batch size was fixed at 1, with model parameters rebuilt after each edit. Decoding used a temperature setting of 0.001. To ensure fair comparison, edits were applied to the 7th layer of parameters for ROME, RECT, and UNKE. Evaluation results before and after the '/' represent outputs for original and paraphrased questions, respectively. FC refers to Factual Correctness.

| Method | Semantic Similarity | Lexical Similarity | | | FC | General Ability |
| --- | --- | --- | --- | --- | --- | --- |
| | Bert-Score | Rouge-1 | Rouge-2 | Rouge-L | FactScore | MMLU |
| **LLaMA2-13B-Chat** | | | | | | 30.96 |
| ROME | 71.42 / 69.35 | 48.03 / 41.57 | 27.59 / 20.72 | 44.49 / 39.92 | 21.96 | 29.60 |
| RECT | 73.78 / 70.36 | 48.40 / 42.65 | 28.86 / 21.36 | 45.70 / 40.58 | 23.53 | 29.74 |
| MEMIT | 76.93 / 74.75 | 45.19 / 43.51 | 26.97 / 22.63 | 41.36 / 38.71 | 26.47 | 29.78 |
| AlphaEdit | 77.14 / 73.39 | 45.38 / 41.09 | 26.05 / 20.64 | 42.50 / 38.92 | 28.08 | 30.55 |
| UNKE | 96.92 / 90.16 | 98.29 / 79.38 | 97.77 / 71.96 | 96.78 / 77.14 | 42.53 | 30.16 |
| AnyEdit | 97.46 / 90.93 | 98.04 / 79.25 | 97.99 / 72.13 | 96.87 / 79.12 | 45.53 | 30.44 |
| PIPE | **97.85 / 91.88** | **99.31 / 85.05** | **98.56 / 77.59** | **98.24 / 83.61** | **48.81** | 30.63 |

best results. As TANGLESCORE increases, editing performance drops significantly. PIPE maintains the smallest decline and its performance improvement over other editing methods is much larger than the average improvement across all samples.

Table 11: The comparison of performance on UNKE of different methods across varying difficulty ranges using LLaMA3-8B, with TangleScore normalized to the [0,1] range using min-max scaling.

| TANGLESCORE | (0, 0.25] | (0.25, 0.5] | (0.5, 0.75] | (0.75, 1] |
| --- | --- | --- | --- | --- |
| Method | Bert-Score | | | |
| ROME | 97.48 / 96.87 | 77.14 / 76.39 | 55.18 / 54.68 | 35.12 / 34.36 |
| RECT | 97.71 / 97.05 | 77.62 / 78.08 | 55.65 / 55.16 | 36.29 / 35.64 |
| MEMIT | 98.03 / 97.31 | 80.48 / 80.12 | 57.74 / 57.31 | 38.40 / 37.70 |
| AlphaEdit | 98.19 / 97.52 | 79.51 / 79.06 | 58.83 / 58.42 | 39.84 / 39.16 |
| UNKE | 98.52 / 97.83 | 97.48 / 96.06 | 67.90 / 67.50 | 45.37 / 44.78 |
| AnyEdit | 98.83 / 98.15 | 96.97 / 96.53 | 72.67 / 70.21 | 48.40 / 47.76 |
| PIPE | **99.07 / 98.31** | **98.21 / 97.69** | **80.53 / 79.10** | **65.35 / 63.73** |

Table 12: Comparison of average editing time per sample (seconds) with different methods

| Method | ROME | MEMIT | RECT | AlphaEdit | UNKE | PIPE |
|---|---|---|---|---|---|---|
| **Time(s)** | 12.09 | 16.16 | 12.88 | 18.53 | 23.56 | 21.32 |

Table 13: Comparison of different batch sizes. We conducted experiments on PIPE using the LLaMA3-8B-Instruct model, with the decoding temperature set to 0.001. Evaluation results before and after the '/' represent outputs for original and paraphrased questions, respectively. FC refers to Factual Correctness.

| Batch Size | Semantic Similarity | Lexical Similarity | | | FC | General Ability |
|---|---|---|---|---|---|---|
| | Bert-Score | Rouge-1 | Rouge-2 | Rouge-L | FactScore | MMLU |
| $2^0$ | 98.64 / 91.71 | 98.48 / 85.42 | 98.89 / 78.03 | 98.44 / 84.07 | 50.91 | 29.51 |
| $2^1$ | 98.62 / 90.32 | 98.32 / 82.82 | 98.67 / 76.45 | 98.57 / 83.78 | 50.27 | 29.28 |
| $2^2$ | 98.53 / 89.58 | 98.18 / 81.90 | 98.88 / 73.82 | 98.33 / 81.39 | 50.16 | 29.11 |
| $2^3$ | 98.37 / 87.44 | 98.25 / 78.56 | 98.32 / 72.48 | 98.04 / 78.68 | 49.80 | 29.35 |
| $2^4$ | 98.40 / 85.74 | 98.21 / 75.33 | 98.47 / 70.89 | 98.73 / 76.12 | 49.81 | 29.37 |
| $2^5$ | 98.09 / 85.16 | 98.18 / 73.97 | 96.11 / 69.16 | 98.51 / 75.86 | 50.13 | 29.63 |
| $2^6$ | 98.18 / 85.42 | 98.07 / 72.65 | 98.56 / 68.68 | 98.80 / 75.69 | 49.45 | 29.54 |

## F.4 TIME COSTS

We evaluated the runtime of various methods on UNKEBench, following the official UNKE protocol. The results are presented in Table 12. All experiments were conducted on a single A800-80G GPU. Due to the inherent complexity of unstructured knowledge editing, our method requires more editing time than structured editing approaches. However, this increased cost is accompanied by a substantial performance improvement. Furthermore, compared to structured editing methods, our approach demonstrates clear advantages in both efficiency and effectiveness.

## G ROBUSTNESS ANALYSIS ON BATCH EDITING AND SEQUENTIAL EDITING

To evaluate the robustness of PIPE in unstructured knowledge editing, we examine both its batch and sequential editing performance on the UNKEBench dataset. In the batch editing evaluation, as shown in Table 13, the model demonstrates stable performance on unstructured editing tasks as the batch size increases, indicating the robustness of PIPE in handling batch edits. For sequential editing, as presented in Figure 13, the performance of all methods declines as the number of edits increases, as expected. However, PIPE demonstrates greater stability compared to other baselines, particularly with a large number of edits. These results affirm the effectiveness of PIPE across both batch and sequential editing, emphasizing its promise as a robust method for unstructured knowledge editing.

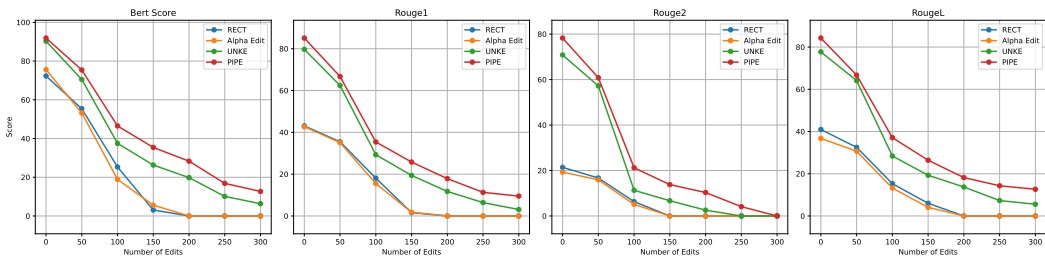

Figure 13: Performance in sequential editing using LLaMA3-8B-Instruct model. We select the first 300 samples in the UNKEBench dataset for sequential editing experiments.

## H CASE ANALYSIS OF RECT, UNKE AND PIPE

Figure 14 presents the generation results of four different methods: RECT, AlphaEdit, UNKE and PIPE. The local key-value editing approaches, RECT and AlphaEdit, exhibit limited effectiveness

Figure 14: The editing targets and their corresponding desired answers, along with the model's outputs after editing using different methods. "Origin" refers to the model's response to the original question, while "Para" refers to its response to the paraphrased question.

in handling complex unstructured knowledge editing tasks. These methods tend to memorize only fragments of the target knowledge and struggle to reproduce the complete editing objective. In comparison, UNKE demonstrates a better ability to convey the intended edits. However, its edited model still fails to fully comprehend the knowledge when answering rewritten questions. In contrast, PIPE shows a stronger grasp of the target knowledge, enabling it to respond accurately to rewritten questions and clearly reflect the intended editing outcome.

## I  BROADER IMPACT

This research focuses on a key aspect of LLMs: reducing hallucinations through knowledge editing to improve human-AI interactions and promote the development of safer, more accountable AI systems. Our findings deepen the scientific understanding of both the capabilities and limitations of current editing techniques, while also drawing attention to the challenges posed by unstructured knowledge editing. These insights not only advance current knowledge but also pave the way for future studies on unstructured knowledge within language models. Additionally, our work underscores the importance of responsible and transparent methodologies, encouraging continued engagement in the broader conversation on ethical AI development. Such emphasis is essential for enhancing the reliability and trustworthiness of language models in practical use.

Despite these advancements, potential societal risks remain. Unstructured knowledge editing of LLMs could be misused to enhance the spread of disinformation. For instance, malicious actors might

leverage this capability to engineer AI systems that recall and reinforce false or biased narratives in a way that appears more coherent and convincing. Such misuse could significantly amplify the reach of fake news and disinformation campaigns, threatening public trust and democratic integrity.

To address these risks, several precautionary strategies can be adopted. First, implementing strict access controls and audit trails is essential to monitor the use of unstructured knowledge editing tools. Second, developers should embed reliable safeguards, such as fact-checking algorithms and transparent output mechanisms, to curb the spread of misinformation. Finally, ongoing monitoring and routine model updates are necessary to prevent the retention of outdated or harmful content. Together, these measures help ensure that the benefits of unstructured knowledge editing are realized without compromising ethical standards or public trust.

## J  ETHICS STATEMENT

This work focuses on improving knowledge editing in large language models (LLMs), a task that has potential implications for model transparency, adaptability, and factual correctness. All experiments were conducted on publicly available models and datasets. No private or sensitive user data was used. We acknowledge that model editing can influence model behavior in significant ways, including the potential to alter safety-related outputs. Therefore, we emphasize that editing methods like PIPE should be deployed with caution in real-world systems and coupled with appropriate safeguards to prevent misuse, such as unauthorized knowledge tampering.

We employed a large language model to aid in the refinement of the English writing throughout the preparation of this manuscript.

