# OpenReview forum: "TangleScore: Tangle-Guided Purge and Imprint for Unstructured Knowledge Editing"
_ICLR.cc/2026/Conference — ICLR 2026 Poster_

### Official Review · Reviewer_bDZe · 2025-10-20

**Soundness:** 2
**Presentation:** 3
**Contribution:** 3
**Rating:** 4
**Confidence:** 4

**Summary:**

The paper addresses the challenge of unstructured knowledge editing in large language models, where existing methods often fail to generalize due to varying instance-specific editing difficulty. It introduces TangleScore, a metric that quantifies how strongly a fact is entangled with a model’s prior knowledge by combining internal representation shifts and semantic gaps between pre- and post-edit outputs. Building on this, the authors propose PIPE (Purge-Imprint Patch Editing), a two-stage adaptive framework that first purges outdated knowledge at an intensity proportional to the TangleScore and then imprints new knowledge using a dynamic loss balancing learning and stability retention. Experiments on UNKEBench, AKEW, and KEBench with LLaMA-2/3 and Qwen2.5 models show that PIPE improves generalization while preserving factual correctness and general abilities. Overall, the work provides both a diagnostic tool and an adaptive editing framework for more robust and interpretable unstructured knowledge editing.

**Strengths:**

The paper addresses the highly relevant problem of editing unstructured knowledge in large language models. It is original in formulating editing difficulty as a quantifiable property of individual knowledge instances. The proposed TangleScore represents a clear conceptual advance by linking internal model entanglement to empirical editing outcomes. Building on this insight, the PIPE framework introduces an adaptive editing strategy that dynamically scales intervention strength based on the measured difficulty of each instance.

The experimental evaluation is thorough, covering multiple model families (LLaMA-2/3, Qwen-2.5) and diverse datasets (UNKEBench, AKEW, KEBench). It includes ablations, structured versus unstructured comparisons, and assessments of general ability retention using MMLU.

Overall, TangleScore provides a valuable diagnostic tool for analyzing editability across models, while PIPE achieves consistent performance gains on the key challenge of generalizing unstructured knowledge edits. Together, they constitute a meaningful conceptual and practical advance in understanding and improving knowledge editing in LLMs.

**Weaknesses:**

While the paper presents an interesting and well-motivated approach, several aspects should be improved to strengthen its conceptual clarity and empirical rigor.

First, the derivation of the TangleScore is insufficiently detailed and difficult to reproduce. It remains unclear how the hidden-layer representations used in the metric are obtained, whether they come from specific transformer layers, averaged across layers, or taken from a particular token position. Similarly, the definition of the answer distributions used in the Sinkhorn distance is vague. The paper does not specify whether these distributions are token logits, embedding vectors, or contextualized sentence representations. This lack of precision makes it difficult to interpret what the TangleScore is actually measuring.

Second, the form of the TangleScore function lacks theoretical justification. The authors divide the internal representation distance by the distance of the answer distributions, but no clear motivation or intuition is provided for this specific ratio. Ablation experiments testing alternative formulations, such as using only one of the two components or other combinations (e.g., $\frac{1}{\text{answer distance}}$ alone), would help validate this design choice and clarify the underlying rationale.

Third, there is a conceptual inconsistency in Section 3.2. The authors first claim that the TangleScore is “_solely determined by the knowledge to be edited_” suggesting it is independent of the model, yet later state that the TangleScore depends on the model’s internal representations. This contradiction raises questions about whether TangleScore truly captures an intrinsic property of the knowledge instance or merely reflects model-specific behavior. A more precise discussion and controlled experiments varying model checkpoints could resolve this ambiguity.

Fourth, the definition of “generalization” in the experiments is unclear. The paper reports improvements in generalization metrics but never explains what type of generalization is tested. (e.g., paraphrase generalization, compositional generalization, or reasoning generalization). Even if the benchmarks (UNKEBench, AKEW) provide paraphrased samples, the authors should explicitly describe their structure and what constitutes success. Moreover, evaluating only a single type of generalization offers a limited view of editing robustness. The brief mention of a “multi-hop comprehension task” in Section 5.2 is particularly confusing, as it is unclear what data or setup is used. Clarifying and expanding these evaluations to include more challenging forms of generalization (e.g., multi-hop or reasoning-based edits) would significantly strengthen the empirical claims.

Finally, while the proposed two-stage editing design (purge then imprint) is conceptually appealing, the evaluation does not isolate or verify the effectiveness of the purge step. All reported results focus on the final edited performance, leaving open whether the purge function truly removes the targeted knowledge before re-imprinting. An explicit evaluation (e.g., measuring model responses after the purge stage but before imprinting) would provide valuable evidence that the method actually unlearns the targeted knowledge.

**Questions:**

* TangleScore derivation: Could the authors clarify how the hidden-layer representations and answer distributions used in TangleScore are obtained?
* TangleScore formulation: What is the motivation for dividing the hidden-state distance by the answer-distribution distance, and have alternative formulations been tested?
* Generalization definition: What type of generalization is evaluated (e.g., paraphrase, reasoning, multi-hop), and how are these samples constructed in the benchmarks?
* Multi-hop task: Could the authors clarify what the mentioned “multi-hop comprehension task” in Section 5.2 refers to?
* Effectiveness of purge step: Have the authors measured the model’s behavior after the purge phase but before imprinting to confirm that the purge truly unlearns the targeted knowledge?
* Relation to unlearning: If the imprint step were removed, would it be possible to employ PIPE as a standalone unlearning method?

---

> ### Author Response · Authors · 2025-11-18
>
> We greatly thank the reviewer bDZe for his/her helpful and insightful comments. We provide our responses to the comments as follows.
>
> 1. Re Weakness 1
>
> Thank you for pointing this out. In Appendix B, we provide examples illustrating how we construct the old–new answer sample pairs. The hidden-layer representations used in TangleScore are obtained from the transformer's down-projection layer (in our experiments, we selected the 7th layer), while the Sinkhorn distance is computed on the embeddings of the old and new answers, which are completely model-agnostic. By taking the ratio of these two quantities, we aim to explore “how strongly the model internally responds when old knowledge changes by a unit amount to new knowledge,” thereby reflecting the model’s inertia toward an old–new knowledge pair.
>
> 2. Re Weakness 2
>
> We define TangleScore as a ratio that measures how much the model’s internal representations change for each unit of output modification, that is, from its original knowledge to the newly introduced one. In this ratio, the numerator represents the degree of internal representation shift, while the denominator quantifies the corresponding output variation. Hence, TangleScore captures the extent of internal adaptation required to produce a given output change, effectively reflecting the model’s resistance or inertia toward editing. The absence of either the numerator or the denominator would make it difficult for TangleScore to fulfill its intended purpose.
>
> 3. Re Weakness 3
>
> We apologize for the misunderstanding. In fact, our intention in the paper was not to claim that TangleScore is entirely determined by the knowledge to be edited, but rather that for a given model, TangleScore is determined by the knowledge to be edited and is independent of the editing method (lines 213–214). More detailed results are presented in Figure 2 and Appendix C.
>
> 4. Re Weakness 4
>
> The generalization mentioned in our evaluation metrics refers to paraphrase generalization. Specifically, we test whether the edited model can provide consistent answers to paraphrased versions of the original questions. Its performance is measured using lexical similarity and semantic similarity. The multi-hop comprehension task refers to the ability of the edited model not only to answer the original question but also to correctly handle derived sub-questions, producing answers consistent with the reference sub-answers in the dataset. We evaluate its performance using factual correctness. We discuss the detailed computation of all these metrics in Appendix F.3. Additionally, we provide the detailed structure of the samples, which has been added to the appendix in the revised version of the paper.
>
> 5. Re Weakness 5
>
> The issue you raised is indeed insightful. Our current work primarily focuses on the introduction of the TangleScore metric and the final performance of the editing method based on it, which is why most experiments are centered on these aspects. In fact, our paper targets unstructured knowledge editing, for which there is no ground truth available as in structured knowledge datasets, making it difficult to directly evaluate the forgetting effect of the model after the purge stage. Nevertheless, in Section 5.4, we conducted an alternative analysis, and the results (Figure 5b) show that, compared with the previous method (UNKE), our approach significantly reduces the model’s tendency to rely on old knowledge. Furthermore, in Appendix E.2, we performed an ablation study on the loss functions governing the two stages. We believe these results provide evidence supporting the effectiveness of the purge step.
>
> 6. Re Q1 - Q5
>
> We notice that these questions are based on the issues raised in the previous Weaknesses section, and we hope that our earlier discussions have addressed your concerns.
>
> 7. Re Q6
>
> We believe that even without the imprint stage, PIPE can function as a standalone unlearning method. As mentioned in our paper (lines 313–316), traditional unlearning methods may suffer from gradient explosion problems, whereas the purge function in PIPE effectively avoids this issue. Moreover, during the purge process, PIPE leverages TangleScore as guidance to adaptively determine the unlearning intensity, achieving better forgetting performance while minimizing side effects.
>
> We hope our response addresses your concerns, and we look forward to further discussion with you.

---

> > ### Comment · Reviewer_bDZe · 2025-11-22
> >
> > Thank you for the detailed rebuttal and for taking the time to address my comments. I appreciate the clarifications provided and have the following additional remarks and suggestions:
> > 1) Thank you for clarifying how hidden-layer representations and answer embeddings are obtained. I strongly encourage adding this clarification to Section 3.1 of the paper. However, several questions remain: Why is the 7th layer chosen specifically? Does this layer vary across model families or sizes? How are the embeddings of old and new answers computed, and which embedding model is used? Clarifying these points would greatly improve reproducibility.
> > 2) With the additional explanation, I now understand the intuition behind the TangleScore ratio. Nonetheless, the paper would still benefit from a small ablation demonstrating the necessity of both numerator and denominator, especially given the authors’ claim that removing either would undermine the metric’s purpose.
> > 3) Thank you for explaining the intended meaning of lines 213–214. I recommend rephrasing this part of Section 3.2 to avoid confusion about whether TangleScore reflects an intrinsic property of the knowledge or a model-specific quantity.
> > 4) I appreciate the clarification regarding paraphrase and multi-hop generalization. I encourage the authors to integrate a brief summary of Appendix F.3 into the main paper for better conceptual clarity.
> > 5) I understand the difficulty of evaluating unlearning directly in an unstructured setting. For a potential resubmission (in case the paper is not accepted), I would recommend adding structured unlearning experiments to more explicitly isolate the effectiveness of the purge step. A method that supports both unlearning and editing would be a valuable contribution.
> >
> > I thank the authors again for the thoughtful rebuttal. However, some concerns remain insufficiently addressed for me to revise my score at this stage.

---

> > > ### Author Response · Authors · 2025-11-23
> > >
> > > Thank you for your thoughtful feedback. We are glad to clarify several of your concerns.
> > >
> > > 1. We selected the 7th layer because we follow prior work [1]. Experiments have shown that when relatively shallow layers (e.g., Layers 5–10) are chosen, the edited model maintains better stability. In contrast, when deeper layers are used, the target information encoded in the key vectors becomes more entangled and therefore more difficult to edit. Since prior work consistently used the 7th layer for all model editing experiments, we adopt the same setting to enable a fair comparison. Regarding the embeddings of old and new answers, we did not apply any special processing; we simply use the model’s tokenizer to obtain standard token embeddings, keeping the procedure consistent with the regular inference process. We will add further explanation in Section 3.1.
> > >
> > > 2. Based on our definition of the TangleScore ratio, the ablation experiment you suggested is unfortunately not feasible. As explained earlier, TangleScore measures the model’s response gap under unit changes from old to new knowledge. This ratio is essential to preserve the invariance properties described in our paper and is key to measuring editing difficulty. If either the numerator or denominator were used alone in place of the complete TangleScore, these properties would be entirely lost, and using such incomplete metrics to guide the purge rate during editing would no longer be meaningful.
> > >
> > > 3. We are glad to resolve this misunderstanding. In the revised manuscript, we will reorganize the relevant section to avoid similar confusion and make our core motivation clearer.
> > >
> > > 4. Thank you very much for the suggestion. We briefly mention the related evaluation metrics in Lines 399–402 of the main paper. Due to space limitations, the detailed explanations were moved to the appendix. In the revised version, we will improve this part by adding a concise summary of the appendix content in the main text.
> > >
> > > 5. Thank you for your understanding. In our future work, we plan to independently apply the forgetting module of PIPE to several standard unlearning datasets [2][3] to further validate the effectiveness of the forgetting step alone.
> > >
> > > We hope the above responses address your concerns, and we look forward to continued discussions with you.
> > >
> > > [1] Everything is Editable: Extend Knowledge Editing to Unstructured Data in Large Language Models
> > >
> > > [2] TOFU: A Task of Fictitious Unlearning for LLMs
> > >
> > > [3] PKU-SafeRLHF: Towards Multi-Level Safety Alignment for LLMs with Human Preference

---

> > > > ### Comment · Reviewer_bDZe · 2025-11-23
> > > >
> > > > Thank you for the further clarifications. I appreciate the additional details regarding the choice of the 7th layer as well as your intention to expand Section 3.1 in the revised manuscript. I understand the motivation to follow prior work for comparability. That said, I would caution that relying on inherited design choices without ablating them can propagate suboptimal or poorly understood assumptions across papers. While I recognize that a full layer-selection ablation may be computationally expensive, even a small-scale version would strengthen the empirical grounding of the method.
> > > >
> > > > Regarding the TangleScore formulation, I respectfully disagree that the suggested ablation is infeasible. I agree that using only the numerator or only the denominator is unlikely to perform well, but demonstrating this empirically would reinforce the claim and make the motivation more robust. Even a brief experiment on a small subset would likely be sufficient and would enhance the empirical justification of the ratio-based design.
> > > >
> > > > Overall, I find the core idea of the paper interesting and promising. My main concern remains the vagueness of the initial submission regarding the motivation and precise computation of TangleScore. Once the authors have incorporated the promised revisions and added the missing clarifications, I would be happy to revisit the updated paper and evaluate whether these changes justify an improved score.
> > > >
> > > > Thank you again for the thoughtful rebuttal.

---

> > > > > ### Author Response · Authors · 2025-11-24
> > > > >
> > > > > Thank you for your further response and for recognizing our idea. In fact, after receiving your previous comments, we have already conducted the relevant experiments. To address your concerns, we have improved the discussions on the TangleScore formulation and layer selection in Sections 3.1 and 4.3, and detailed experimental results are reported in Appendices E.1 and E.3. We have uploaded the revised manuscript, and we sincerely hope that these additional explanations and experiments can address your questions and further substantiate the validity of our work.
> > > > >
> > > > > We would be very happy to answer any further questions you may have or to receive any additional feedback on the revised manuscript.

---

> > > > > > ### Comment · Reviewer_bDZe · 2025-11-24
> > > > > >
> > > > > > Thank you for the thorough revision and for addressing all of my earlier questions and concerns. The added explanations and new ablation studies significantly improve the clarity and empirical grounding of the paper. I appreciate the effort put into the updated manuscript, and I am accordingly updating my score.

---

> > > > > > > ### Author Response · Authors · 2025-11-25
> > > > > > >
> > > > > > > Thank you for your recognition of our work and for your thoughtful discussions and suggestions.
> > > > > > >
> > > > > > > If you have any further questions or comments, we would be glad to continue the conversation with you.

---

### Official Review · Reviewer_zBPp · 2025-10-29

**Soundness:** 2
**Presentation:** 2
**Contribution:** 2
**Rating:** 4
**Confidence:** 4

**Summary:**

This paper addresses the generalization of edits for unstructured knowledge. The authors propose a metric, TANGLESCORE, to quantify the intrinsic difficulty of editing a specific knowledge instance. This metric is based on the internal representation shift and the output semantic gap between the old and new knowledge. Leveraging this insight, they introduce PIPE, a framework that adaptively modulates its editing strategy based on the TANGLESCORE. Experiments across four LLMs and three datasets show that PIPE outperforms baselines, particularly in generalization on paraphrased prompts and factual correctness.

**Strengths:**

1. The paper introduces the TANGLESCORE and diagnoses why unstructured KE fails. The conceptualization of editing difficulty as an intrinsic, quantifiable property of the model-knowledge is a valuable contribution.
2. The PIPE method is a direct and logical consequence of the TANGLESCORE analysis. The two-stage "purge-then-imprint" paradigm is well-justified.
3. The authors evaluate on multiple modern, open-source LLMs. The use of multiple datasets and the inclusion of structured KE demonstrates the method's applicability.

**Weaknesses:**

1.	The definition of TANGLESCORE as a ratio between internal representation shift and output semantic gap is not fully justified. Why this specific formulation (a ratio)? Why greater semantic gap of outputs causes smaller TangleScore? The paper would be stronger with a more rigorous theoretical or empirical justification for this specific choice.
2.	The PR definition is unclear in eq3 and fig4. As stated “where γ = log λ_max/ log λ_min is the scaling factor, and λ_max and λ_min represent the maximum and minimum values,” what is the maximum and minimum for? Why need two hyper-parameter to set this γ? The rationale for these specific values is attributed only to "preliminary exploratory experiments." Furthermore, λ_min only appears once, used to define the scaling factor γ, making its role ambiguous and the overall formulation seem arbitrary. This lack of clear justification for these critical, hard-coded values undermines the method's robustness.
3.	The claim that TANGLESCORE is an "intrinsic property" is a strong one, but the analysis is based on showing the distribution remains similar pre- and post-edit. This doesn't necessarily mean the score for individual samples is unchanged.
4.	Confusing and contradictory analysis in fig.5(b). The analysis in Section 5.4, meant to show PIPE's effectiveness, is poorly presented and potentially contradictory. The method is under-explained. The Y-axis of Figure 5(b) is unlabeled (presumably 'Density'). More critically, the X-axis is labeled 'Probability(0.0-1.0),' while the text states 'negative log-probability' was used. These two metrics are inversely related. If the axis is 'Probability,' the observed rightward shift for PIPE would imply a higher probability of outputting old knowledge, directly contradicting the paper's claim of suppression. This ambiguity makes the entire analysis unverifiable.
5.	The paper's central thesis is that existing methods fail on high-TangleScore samples and that PIPE is designed to solve this. However, the paper does not present the critical evidence: a plot of PIPE's performance versus TangleScore and improvements on hard knowledge against other methods. Without this, it is impossible to verify if PIPE's improvement comes from solving the "hard sample" problem, or from just being a better method on average.
6.	The paper's mathematical notation is ambiguous and mathematically inconsistent. L_purge (Eq. 5) is explicitly defined for a single 'i-th knowledge item'. In contrast, L_consistency (Eq. 6) and L_learn (Eq. 8) are defined as summations over the entire batch. The final objective in Eq. 10 then adds these mismatched terms. This sloppiness makes the precise formulation of the final objective unclear. And the expressions like “current key output for the i-th query” “original key vector” are unclear to show how they are obtained.
7.	The figures (fig.1 and fig.4) are blurry and fig.4 shows traces of post-processing of changing some characters to “TS”. Fig.5(b) lacks a legend. Line 292: “higher TANGLESCORE implies hard-to-edit knowledge…while higher TANGLESCORE denotes easier samples”. Line 342: “u denote the numbers of pre-training samples, tokens per input, and edited samples, respectively.” Eq.8 is under-explained.

**Questions:**

1.	Could the authors elaborate on the design choice for the TANGLESCORE ratio? What is the intuition behind dividing the internal shift by the output shift?
2.	The calculation of r_old and r_new relies on a specific prompt-answer template. How sensitive is the TANGLESCORE value to variations in this prompt template?
3.	In Table 8, does the reported Time for PIPE include the pre-computation of TANGLESCORE?
4.	How is the "original key vector" $\tilde{k}_{i}^{orig}$ used in the knowledge purge function (Eq. 5) obtained?

---

> ### Author Response · Authors · 2025-11-18
>
> We greatly thank the reviewer zBPp for his/her helpful and insightful comments. We provide our responses to the comments as follows.
>
> 1. Re Weakness 1
>
> We formulate TangleScore as a ratio in order to directly measure how much internal representational change is required for the model to transition from its original knowledge to the newly injected knowledge. In this formulation, the numerator quantifies the degree of shift in the model’s internal representations, whereas the denominator captures the corresponding level of change observed at the output side. By taking their ratio, TangleScore expresses the amount of internal adjustment needed to induce a given output modification. Compared with this ratio-based design, alternatives such as weighted sums or simple differences cannot convey the same semantic clarity, nor do they provide an equally interpretable link between internal updates and external behavioral change.
>
> 2. Re Weakness 2
>
> Thank you for pointing this out. In fact, λ_min and λ_max are used solely to constrain PR within a specific range, and the scaling factor in the paper is simply a mathematical expression that fits this range. We need to map TangleScore into a reasonable interval so that it can appropriately guide the forgetting process, and our preliminary experiments showed that mapping it to a range that is too high or too low leads to performance degradation. We discuss the impact of the PR range on editing effectiveness in Appendix E.1.
>
> 3. Re Weakness 3
>
> We agree with your point, and we conducted corresponding experiments in Appendix C.3. We applied multiple editing methods across different models and plotted the scatter of each sample’s TangleScore before and after editing. The results show that the TangleScore of individual samples remains essentially unchanged, which further supports the conclusion presented in the main text. In addition, we performed experiments on both structured and unstructured knowledge, demonstrating that this conclusion holds with good generality.
>
> 4. Re Weakness 4
>
> We apologize for not explaining this clearly. The Y-axis of Figure 5(b) is indeed the KDE-estimated density, and the X-axis represents the negative log-probability, as you noted. Therefore, as the negative log-probability increases, the model’s probability of producing the old answer decreases, which is consistent with the suppression effect we claim. We will correct this and clarify the presentation in the revised version of the paper.
>
> 5. Re Weakness 5
>
> Thank you for the suggestion. For low-TangleScore samples, traditional methods already perform well, leaving limited room for PIPE to improve. In contrast, as discussed in Section 3.2, traditional methods degrade substantially on high-TangleScore samples. Thus, PIPE’s gains in average performance mainly stem from its superior effectiveness on these difficult cases. To make this clearer, we added experiments that edit only high-TangleScore samples and compared PIPE against other methods. We found that PIPE’s advantage is even larger on this subset than on the full dataset, confirming that its contribution to average performance is driven primarily by high-TangleScore samples. We will elaborate on these results in the revised manuscript.
>
> 6. Re Weakness 6
>
> We apologize for the ambiguity. In our method, all loss functions are defined over the full batch; we will correct the miswritten formulas in the revision. The key vectors are taken from the hidden representation of the last token at a Transformer layer, and the new key vector is obtained by adding a residual to the original so that the model’s output matches the target answer. We briefly mentioned this in lines 324–328 and will provide a clearer explanation in the revised version.
>
> 7. Re Weakness 7
>
> Thank you for pointing this out. We apologize for the blurriness in the figures. We will still do our best to make the figures clearer in the revised version. As for Eq. 8, it is essentially an L2 loss whose purpose is to make the new key vector close to the desired key vector. We will also carefully check and correct the related typos in the revision. We are sorry for the inconvenience this may have caused in your reading experience.
>
> 8. Re Q1 and Q4
>
> We notice that these questions are based on the issues raised in the previous Weaknesses section, and we hope that our earlier discussions have addressed your concerns.
>
> 9.  Re Q2
>
> Thank you for raising this question. In fact, we do not use any specific prompt–answer template. As mentioned in Appendix B (Table 3), our input is simply constructed by concatenating the dataset’s prompt (i.e., the question itself) with the old and new answers. Therefore, there is no issue of choosing a particular prompt–answer template.
>
> 10. Re Q3
>
> The time cost we report for PIPE is end-to-end, including both the computation of TangleScore and the editing process.
>
> We hope our response addresses your concerns, and we look forward to further discussion with you.

---

> > ### Author Response · Authors · 2025-11-24
> >
> > Thanks again for your helpful feedback. Just following up to check if we've addressed your concerns and if you have any further questions.

---

### Official Review · Reviewer_76iD · 2025-10-30

**Soundness:** 3
**Presentation:** 3
**Contribution:** 3
**Rating:** 6
**Confidence:** 2

**Summary:**

This paper proposes TangleScore, a metric intended to quantify the intrinsic difficulty of editing a given knowledge in an LLM, and a corresponding editing framework, PIPE, whose purge and imprint strengths are adaptively modulated by the proposed TangleScore. Specifically, TangleScore measures (i) the internal representation shift between model states associated with old vs. new knowledge and (ii) the semantic gap between answers before/after editing. Empirically, the authors show that higher TangleScore correlates with worse post-edit generalization on paraphrases. Based on this, PIPE applies stronger purging (unlearning) and more assertive imprinting on high-TangleScore (hard) instances using a gradient-bounded purge loss and a consistency-regularized imprint loss with an instance-wise weight. Experiments on UNKEBench, AKEW and KEBench across LLaMA and Qwen-family models seem to indicate that PIPE improves generalization over ROME/MEMIT/RECT/AlphaEdit and recent unstructured editors.

**Strengths:**

+ Overall I feel that tangleScore gives a practical, per-instance signal for how “entangled” an edit is, and PIPE uses it to adapt purge/imprint strength, which indeed turns a vague notion (e.g., “some edits are harder”) into an actionable control knob.

+ Empirically, by introducing stronger purging (unlearning) and more assertive imprinting on high-TangleScore (hard) instances using a gradient-bounded purge loss and a consistency-regularized imprint loss, the method improves generalization on unstructured edits while preserving performance on structured edits and holding up model-wide abilities.

**Weaknesses:**

- The specific TangleScore form (choice of representation/output distances and ratio) feels ad-hoc and lack in-depth theoretical justification.

-  Computing the score and running a two-stage edit add overhead, and it would be better if the authors can provide clearer cost–quality tradeoffs (runtime, memory, throughput) needed for large edit streams.

**Questions:**

Please refer to my summery of weaknesses.

---

> ### Author Response · Authors · 2025-11-18
>
> We greatly thank the reviewer 76iD for his/her helpful and insightful comments. We provide our responses to the comments as follows.
>
> 1. Re Weakness 1
>
> Thank you for the comment. While TangleScore，using cosine distance for hidden representations, Sinkhorn distance for outputs, and their ratio，may seem heuristic, its design is guided by both empirical observations and conceptual alignment with quantifying editing resistance: the cosine distance captures shifts in internal semantic space, the Sinkhorn distance measures output changes, and their ratio reflects internal adjustment per unit of behavioral change. As shown in Section 3.2 and Appendix C, TangleScore consistently correlates with editing difficulty and generalization, supporting this formulation.
>
> 2. Re Weakness 2
>
> The issue you raised is highly relevant. In Appendix G.3, we provide detailed reports on the **time and memory costs** of PIPE for large-scale editing, along with comparisons to other methods. The results indicate that, owing to the intrinsic complexity of unstructured knowledge editing, our method requires more editing time than structured editing approaches. Nonetheless, this comes with a substantial performance gain. Additionally, relative to structured methods, our approach shows distinct advantages in both efficiency and effectiveness.
>
> We hope our response addresses your concerns, and we look forward to further discussion with you.

---

> > ### Author Response · Authors · 2025-11-24
> >
> > Thanks again for your helpful feedback. Just following up to check if we've addressed your concerns and if you have any further questions.

---

### Official Review · Reviewer_7HRu · 2025-10-31

**Soundness:** 2
**Presentation:** 3
**Contribution:** 2
**Rating:** 4
**Confidence:** 3

**Summary:**

This paper addresses the challenge of unstructured knowledge editing for LLMs, where existing methods often fail to generalize. To address this, the paper first proposes TANGLESCORE, a novel metric designed to quantify the difficulty of editing specific knowledge. It is calculated based on the shift in the model's internal representations and the semantic gap between the old and new output answers. The authors demonstrate that a higher TANGLESCORE correlates with poorer generalization performance in existing editing methods. To achieve editing, they propose PIPE, a new two-stage editing framework that leverages TANGLESCORE to adapt its strategy. It first employs a purge phase making model unlearn the outdated information, adjusted based on the TANGLESCORE. This is followed by an imprint phase that carefully incorporates the new knowledge while preserving the model's general capabilities. Through extensive experiments on multiple LLMs and benchmarks, the authors show that PIPE outperforms SOTA methods, especially in generalization.

**Strengths:**

1. The paper targets an important and practical challenge of editing unstructured, free-form knowledge, which is more complex than editing simple factual triplets.

2. The proposed two-stage "purge-and-imprint" framework is intuitive, and separating the process of forgetting old information from learning new information is a promising direction.

**Weaknesses:**

1. The TANGLESCORE metric is defined using model outputs and internal states after an edit has been performed. However, the PIPE method claims to use TANGLESCORE to guide the edit itself. This is a circular dependency that the metric required to perform the edit can only be computed after the edit is complete, leaving a fundamental gap in the proposed methodology.

2. The paper’s core purge-then-imprint framework has been explored in prior work. For instance, [1] also employs a similar two-stage process of first erasing outdated knowledge before introducing new facts. The current paper fails to discuss such relevant work, which overstates the novelty of its proposed framework.

3. The paper also fails to discuss existing approaches quantifying editing difficulty. For instance, [2] demonstrated that the perplexity can serve as a strong indicator of editing difficulty. The authors do not mention this or other potential baseline metrics. This omission weakens the novelty and thoroughness of the proposed metric.

[1] Enhancing Multi-hop Reasoning through Knowledge in Large Language Model Editing.

[2] The Butterfly Effect of Model Editing: Few Edits Can Trigger Large Language Models Collapse.

**Questions:**

See weaknesses.

---

> ### Author Response · Authors · 2025-11-18
>
> We greatly thank the reviewer 7HRu for his/her helpful and insightful comments. We provide our responses to the comments as follows.
>
> 1. Re Weakness 1
>
> We apologize for the misunderstanding. In fact, the model undergoes **two separate** forward passes. In the first pass we feed the old and new knowledge into the model to obtain the corresponding outputs and compute the TangleScore. At this stage no editing is performed. In the second pass the TangleScore obtained from the first step is used to guide the actual editing process. The metric needed for editing does not rely on the model after editing and can be obtained from a single forward pass, so there is **no circular dependency**.
>
> 2. Re Weakness 2
>
> Thank you for raising this point. First, the KELE paper you mentioned focuses on improving multi-hop reasoning. Its erasure component is designed to quantify residual old knowledge and optimize performance on multi-hop tasks, thereby reducing the model’s tendency to revert to outdated answers. In contrast, our work specifically aims to identify a metric that captures the inertia of old knowledge, and to use this metric to guide the forgetting operation. This allows the model to more effectively remove outdated information before injecting new knowledge. Second, our method targets unstructured knowledge editing, where traditional token-level localization and erasure techniques are no longer applicable. PIPE carries out forgetting and imprinting directly at the behavioral level of the model in an end-to-end manner and does not rely on any predefined structure of the knowledge being edited. KELE, on the other hand, performs erasure and injection within a rank-one editing framework, which is inherently suited for structured, pattern-fixed factual knowledge rather than free-form, unstructured text.
>
> 3.  Re Weakness 3
>
> The perplexity metric mentioned in the referenced paper is indeed an effective indicator for measuring model degradation after editing. However, it focuses solely on monitoring post-edit performance decline and cannot be used to optimize the editing process itself. In contrast, our proposed TangleScore is designed to guide an adaptive editing procedure and directly improve editing outcomes. Moreover, perplexity evaluates whether the model collapses by examining the model’s output distribution as a whole, serving as an indicator of overall model health. TangleScore, on the other hand, quantifies the intrinsic difficulty of transforming old knowledge into new knowledge for a specific edit. It measures the editing difficulty, not the model’s global condition. Thus, the two metrics differ fundamentally in purpose and function. We will include a detailed comparison with the two referenced papers in the Related Work section of the revised manuscript. In addition, we will explore whether the concept of perplexity can be integrated into our proposed method to guide the editing process from additional dimensions.
>
> We hope our response addresses your concerns, and we look forward to further discussion with you.

---

> > ### Author Response · Authors · 2025-11-24
> >
> > Thanks again for your helpful feedback. Just following up to check if we've addressed your concerns and if you have any further questions.

---

### Author Response · Authors · 2025-12-01
**A Summary of Review, Rebuttal, and Discussion**

Dear Area Chair,

We are deeply regretful about the recent information leakage incident within the community. To support a fair, transparent, and comprehensive evaluation, we summarize the reviewers’ comments and our rebuttal. We also note that some reviewer comments contain clear inaccuracies or misunderstandings, and the reviewers did not respond after our clarifications.

Across the reviews, the key contributions of our paper were acknowledged:
1) We propose an effective solution for **unstructured knowledge editing**, a more challenging and realistic setting beyond structured editing.
2) We introduce **TangleScore**, enabling a quantitative measure of editing difficulty.
3) We design a **purge–imprint framework (PIPE)** based on TangleScore, validated across modern open-source LLMs on both structured and unstructured tasks.

During the rebuttal, we addressed every reviewer’s concerns in detail.

Reviewer **bDZe** questioned the editing location, TangleScore formulation, and some ambiguous statements. We revised the paper, added experiments, and clarified all points. The reviewer acknowledged our response and **raised their score from 4 to 6** (24 Nov 2025, before the leak incident).

Reviewer **zBPp** pointed out ambiguous expressions and requested experiments on high-Tangle samples. We clarified the expressions and added the requested results. The reviewer also claimed that the scaling factor and sample-level TangleScore invariance were unclear, though both were **explicitly addressed in the original manuscript** and reiterated in our rebuttal. The reviewer did not respond afterward.

Reviewer **76iD** questioned the TangleScore formulation; we clarified this. They also claimed the method lacked a computing-cost analysis, although such analysis was **already included** in the original paper.

Reviewer **7HRu** argued that our method is infeasible and suffers from a **circular dependency**, a misunderstanding of the clearly described PIPE procedure. They further cited prior work on two-stage methods and perplexity-based difficulty indicators. After reviewing the cited papers, we found that the two-stage method (KELE) is **only applicable to structured and multi-hop tasks**, fundamentally different from our unstructured setting. Moreover, perplexity is **not an editing-difficulty metric**, but a degradation indicator. These make the comparisons inappropriate. We clarified these issues, but the reviewer did not respond. We kindly ask you to consider this context in your assessment.

We sincerely appreciate all reviewers for their feedback and thank you for your time, effort, and service to the community. If further clarification is needed, we would be glad to provide it.

Best regards,

Authors

---

### Meta-Review · Area_Chair_6d6v · 2026-01-06

**Summary:**

The paper addresses the problem of knowledge editing. It proposes the TangleScore metric, which is meant to evaluate how likely a given edit is to succeed. The paper shows that this measure is correlated with actual success of editing. It then goes on to propose an editing method that uses this metric in a two stage procedure that intuitively adjusts the edit strength to how resistive the model is to editing.
Empirical evaluation on unstructured knowledge editing shows that the approach improves editing efficacy.
The reviewers had several concerns: the rationale for the metric was not clear, as well as details of its implementation (the Sinkhorn part is indeed quite vague, and not completely clarified in the author response), lack of discussion of related methods for editing efficacy (e.g., based on perplexity) and knowledge deletion+editing (the authors address these papers in the response, but a more comprehensive comparison acknowledging the conceptual similarities would be better), and issues with presentation.
Given the above issues, the paper can benefit from another round of improvement.

**Reviewer Concerns:**

The concerns of reviewer bDZe were largely addressed, and they mentioned raising their score, which the authors also write they did (authors say it was to 6). The other reviewers were not responsive.
One reviewer had claims about related work, which were addressed by the authors, but I do agree these papers are relevant (especially KELE). Other reviewer had comments about presentation, which are unlikely to have been addressed completely.

**Reviewer Scores:**

One reviewer did change their score so I'm assuming the score for this paper is 4,4,6,6 and it is indeed borderine.

---

### Decision · Program_Chairs · 2026-01-26

Accept (Poster)